# Smart Robotic Strategies and Advice for Stock Trading Using Deep Transformer Reinforcement Learning

**Nadeem Malibari** [1,*,†] , **Iyad Katib** [1,†] and **Rashid Mehmood** [2,†]

1   Department of Computer Science, Faculty of Computing and Information Technology, King Abdulaziz University, Jeddah 21589, Saudi Arabia
2   High Performance Computing Center, King Abdulaziz University, Jeddah 21589, Saudi Arabia
*   Correspondence: nmahimalibari@stu.kau.edu.sa
†   These authors contributed equally to this work.

**Abstract:** The many success stories of reinforcement learning (RL) and deep learning (DL) techniques have raised interest in their use for detecting patterns and generating constant profits from financial markets. In this paper, we combine deep reinforcement learning (DRL) with a transformer network to develop a decision transformer architecture for online trading. We use data from the Saudi Stock Exchange (Tadawul), one of the largest liquid stock exchanges globally. Specifically, we use the indices of four firms: Saudi Telecom Company, Al-Rajihi Banking and Investment, Saudi Electricity Company, and Saudi Basic Industries Corporation. To ensure the robustness and risk management of the proposed model, we consider seven reward functions: the Sortino ratio, cumulative returns, annual volatility, omega, the Calmar ratio, max drawdown, and normal reward without any risk adjustments. Our proposed DRL-based model provided the highest average increase in the net worth of Saudi Telecom Company, Saudi Electricity Company, Saudi Basic Industries Corporation, and Al-Rajihi Banking and Investment at 21.54%, 18.54%, 17%, and 19.36%, respectively. The Sortino ratio, cumulative returns, and annual volatility were found to be the best-performing reward functions. This work makes significant contributions to trading regarding long-term investment and profit goals.

**Keywords:** stock trading; transformer; deep reinforcement learning; machine learning; Tadawul; stocks; robotic advice; robotic strategies

## 1. Introduction

A competitive strategy for trading stocks is critical for investment businesses. It can maximize capital to maximize performance, such as targeted return. Brokers usually estimate the prospective return and risk of stocks to maximize returns; however, due to the complex nature of stock markets, it is difficult for analysts to analyse all relevant elements manually.

A commonly used buzzword in financial technology and asset management is "robotic strategies and advice" (or "Robo-Advice"). This is based on artificial intelligence (AI), and has transformed the business model for financial advisers and wealth managers. It refers to a fast-growing new breed of digital offering that provides investors with personalized investment services through a platform that integrates interactive and intelligent components, rather than making appointments with human advisors [1–3].

Machine learning (ML) and deep learning (DL) approaches can learn trading strategies on the basis of historical data provided for each stock, and can extract more profitable patterns that human traders cannot quickly discover. For an ML or DL model, feature engineering is the act of applying domain knowledge to add more features to the input data. However, with the increasing availability of data, ML techniques have revolutionized and achieved great success across a broad spectrum of academic disciplines and practical scenarios, including medical forecasting, natural language processing (NLP), picture

recognition, and so on. The extraordinary performance of machine learning approaches is based on their capability to uncover complicated non-linear patterns and investigate unstructured links without making prior assumptions. Naturally, the financial world and strategy researchers have closely paid attention to the many theoretical developments and practical application achievements of ML and DL techniques.

Deep reinforcement learning (DRL) is a branch of the ML field that is able to combine two powerful techniques: Data-hungry deep learning (DL) and the older, process-oriented reinforcement learning (RL). Traditional RL involves agents making decisions or interacting in an environment through trial and error, where Markov decision processes (MDP) are used to model the process. By acting in the environment, the RL agent earns a reward, and the goal of this agent is to learn to choose those actions that maximize the expected cumulative reward over time. In other words, by noting the results of the actions that it executes in the environment, the agent tries to learn an optimized sequence of actions to execute in order to reach its goal. Sutton and Barto, in their innovative work [4], laid the foundation for a completely new field that would have a profound effect on neuroscience, as well as ML methods. Generally, RL was used with few data, and its behaviour was quite complex. More recently, however, due to the advent of deep neural networks, RL has gained massive power to take on more complicated problems.

The Saudi Stock Exchange (Tadawul) is among the largest liquid stock exchanges globally, which is the only entity in the Kingdom of Saudi Arabia authorized to act as a Securities Exchange [5]. The Saudi Stock Exchange (Tadawul) achieved a total market capitalization of SAR 12,178 trillion in the week ending 12 May 2022, placing it among the top exchanges in the world [6]. Figure 1 depicts the share of foreign investors in the total market capital, compared to Saudi investors; according to market data, foreign investors owned 3.18% of the total market capitalization. Over 136 companies are listed on the Tadawul. Based on the type of industry they operate in, these companies are divided into 21 categories, each of which has stock financial indices. As an example, Etihad Atheeb Telecommunication, Saudi Telecom, Mobile Telecommunications Company Saudi Arabia, and Etihad Etisalat are all listed under the Telecommunication Services industry and, so, are included in the Telecommunication Services Index (TTSI), which serves as the stock finance index.

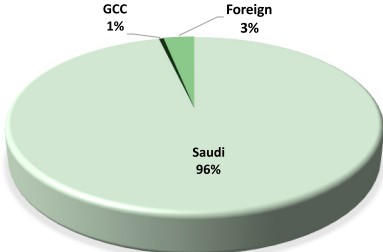

**Figure 1.** Investor Ownership in Saudi Stock Exchange (Tadawul).

### 1.1. Contributions of the Paper

In this research, we integrate modern transformer deep learning (TDL) into a traditional deep reinforcement learning (DRL) architecture to process financial signals and automated trading using data from the Saudi Stock Exchange (Tadawul). The proposed transformer network allows for prediction of the best trading strategy without looking back to track the price movements of stocks. Based on the data of four different industrial indices of the Saudi Stock Exchange (Tadawul)—Saudi Telecom Company (7010), Al-Rajihi Banking and Investment (1120), Saudi Electricity Company (5110), and Saudi Basic Industries Corporation (2010)—the proposed method facilitates the optimal learning of hyper-parameters by automatically selecting the best reward function to increase the net worth and profit on the initial investment. Selection of the best reward function is based on the maximum outcome of the network and return on investment, using a list of seven reward functions; namely, the Sortino ratio, cumulative returns, annual volatility,

omega, Calmar ratio, max drawdown, and normal reward without any risk adjustment. The Sortino ratio reward function provided the highest average increase in net worth for Saudi Telecom Company, Saudi Electricity Company, Saudi Basic Industries Corporation, and Al-Rajihi Banking and Investment, at 21.54%, 18.54%, 17%, and 19.36%, respectively.

The main contributions of our work can be summarised as follows.

- We propose a Transformer DRL-based framework for stock trading. The model does not require a sliding look-back window to track price movements, as it employs a transformer network architecture to pick the best trading policy, which is automatically identified by the intrinsic attention mechanism of the network.
- We enhance our data set by including several widely used and valuable trading technical indicators. With the addition of these technical indicators, we are able to augment our data set with important relevant information, which is well-complemented by the forecasted data calculated from our prediction model. Our proposed model benefits from a good balance of observations using this combination of features.
- The proposed technique optimizes the reward during the training process by integrating risk-adjusted return metrics, including the max drawdown, Sortino ratio, Omega, cumulative returns, annual volatility, Calmar ratio, and a normal reward function without risk adjustment.
- The utilization of various reward functions provides abundant possibilities for exploring the policy space and prohibits the agent from taking an imperfect, but acceptable action.

### 1.2. Outline of the Paper

The remainder of this paper is organized in the following manner. In Section 2, we explain the different approaches to stock trading and how the increasing availability of data has revolutionized ML techniques, which have achieved great success due to their ability to uncover complicated non-linear patterns and investigate unstructured links without making prior assumptions. In Section 3, the methodology and the proposed model are explained, including topics such as the trading environment, state space, action space, the data set employed in this study, processing of data, feature extraction, model architecture, and learning algorithms. In Section 4, we provide details regarding the computational cost, hyper-parameters used, and experiments performed during the training of the model. In Section 5, we discuss the results of our experiments, along with the possible future extensions of our work. Finally, we conclude our work in Section 6.

### 2. Related Work

As an active research field that has appeared recently, generating automated trading stock trading signals based on the financial conditions of stock markets has become a favoured venue among scientists conducting research studies from various perspectives. In general, there are two approaches to stock trading: knowledge-based techniques, in which trading strategies are designed on the basis of financial research, mathematical equations, and/or on the experience of traders; and ML-based techniques, in which strategies are learnt from the available historical data of stock indices [7]. Knowledge-based techniques necessitate human reasoning and the indication of trends in stock data. As a result, they take a lot of time and have flaws such as accuracy, consistency, and imperfections, limiting their ability to generalize financial market strategies [8]. Due to the manual handling of data, the quality of the knowledge-based methods need to be thoroughly examined prior to implementation.

Multi-layer perceptrons (MLPs), hybrid artificial neural networks, and dynamic artificial neural network (DAN2) have been used by Guresen, Kayakutlu, and Daim [9] to forecast the stock index of the NASDA. They described that the MLP model accurately predicted the first movement as down, with a small difference of 0.54% between the actual and predicted realized value (1747.17). They concluded that the MLP model is a useful and powerful tool for the forecasting of stock movements. Long short-term memory (LSTM)

networks can identify long-term dependences and avoid the gradient vanishing problem. They make use of historical data through output, input, and forget gates. Convolutional neural networks (CNNs) are a variant of the multi-layer perceptron (MLP) that excel in pattern recognition, and which have increasingly been used for time-series analysis and prediction. Selvin et al. [10] have implemented a recurrent neural network (RNN), CNN, and LSTM to predict the future prices of stocks for three companies listed on the National Stock Exchange (NSE). They used the sliding window approach to predict the values on a short-term basis. The achieved the percentage error was 3.90% for RNN, 2.36% for CNN, and 4.18% for LSTM. Related work is summarized in Table 1.

**Table 1.** Studies focused on the prediction of stocks using ML, DL, and DRL techniques.

| Reference | Year | Data Set | Model | Application | Results |
|---|---|---|---|---|---|
| Guresen et al. [9] | 2011 | NASDA stock | MLP DAN2 Hybrid ANN | Forecasting of stock indices. | MSE: 0.54% |
| Selvin et al. [10] | 2017 | National Stock Exchange | RNN CNN LSTM | Prediction of future stock price | MSE RNN: 3.90% CNN: 2.36% LSTM: 4.18% |
| Nikou et al. [11] | 2019 | iShares MSCI United Kingdom | ANN, RF, SVR, LSTM | Prediction of closing price of stock | RMSE ANN: 0.45 SVR: 0.34 RNN: 0.38 LSTM: 0.30 |
| Malibari et al. [12] | 2021 | Saudi Stock Exchange | Transformer network | Prediction of closing price of stock | Accuracy over 90% |
| Moody et al. [13] | 1998 | S&P 500 stock index | RRL | Performance check of RRL for trading and profit | Hold strategy: 0.34 Average strategy: 0.84 Voting strategy: 0.83 |
| Xiong et al. [14] | 2018 | Dow Jones 30 stocks | DRL | Prediction of future stock price | Annualized Std. Error of 13.62% Sharpe ratio : 1.79 |
| Gudelek et al. [15] | 2017 | Google finance | 2D-CNN | Prediction of future stock price | 70% accuracy |
| Wang et al. [7] | 2017 | Dow Jones 30 stocks | Portfolio management | deep Q-learning | - |
| Deng et al. [16] | 2017 | Stock IF-contract | Financial Signal Representation and Trading. | DRL | 0.523 PR |
| Luo et al. [17] | 2019 | Stock IF-contract | AI-trader's performance | CNN-DDPG | - |
| Li et al. [18] | 2020 | 10 US equities | single-stock trading strategies | DQN Double DQN Dueling DQN | DQN outperformed the other two techniques |
| Our Work | 2022 | Saudi Stock Market(Tadawul) | 4 stocks trading strategies Robo Advice | Decision Transformer | An increase of around 20% was seen in net worth |

Nikou, Mansourfar, and Bagherzadeh [11] have used four ML and DL models to predict the closing price of the iShares MSCI United Kingdom index. They implemented artificial neural network (ANN), random forest (RF), support vector regression (SVR), and LSTM models for forecasting. The experiment results showed that the LSTM model enhanced the prediction performance of a DL model incorporating the emotional tendencies of investors. Ugur Gudelek, Arda Boluk, and Murat Ozbayoglu [15] have used a 2D CNN for trend identification, where evaluation of model performance yielded accuracy figures of 72%, which is very promising.

The next important step involved a new architecture, Transformer [19], which uses attention mechanisms that harness self-attention processes to analyse whole input series, removing the issues that come with extended sequences. Malibari et al. [12] have proposed

a Transformer neural network for predicting the closing price of Saudi stock exchange for next trading day. Their proposed method used the self-attention mechanism for learning non-linear and dynamic patterns of time-series data of stock indices. They achieved an accuracy of over 90% in accurately predicting the closing price for the next trading day.

RL is an emerging area of artificial intelligence (AI) that has gained much success and acceptance in various applications, including video games, manufacturing, robotics, and aerospace. The concept of RL became prominent with the demonstrated success against human champions in board games such as chess, go, and shogi (Japanese chess). The beginning of RL can be traced back to the 1950s [4] and, since then, it has given rise to a wide variety of fascinating applications in machine control and gaming. It was not until 2013, when DeepMind researchers exhibited their application in Atari games, that it outperformed humans in the majority of games [20]; in particular, Mnih et al. [21] have released a paper, in 2015, demonstrating that a computer taught by Google's DeepMind team to play seven different Atari games, and defeated the best human players in three of the games. After two years, they enhanced their model, employing it to play a total of 49 different games, half of which performed beyond the human difficulty level [22]. Moody et al. [13] were the first to apply RL to financial transactions, proposing a recurrent reinforcement learning (RRL) trading system in which the price is fed directly into the model as learning features for training. Zarkias et al. [23] have established a unique price-tracking approach to choose successful judgments, by re-structuring trading as a control issue and learning trading methods based on the following of trends. They concluded that the performance of a trading system (i.e., the buy and hold strategy) is best when using the accumulated wealth and Sharpe ratio as reward functions, and that it provides less risky results, when compared to maximum drawdown.

With the introduction of deep reinforcement learning, neural networks have brought about a revolution in the field of RL—just as they did in every other area of artificial intelligence research. Better performances and rich feature extraction can be attained through the combination of deep neural networks and RL, yielding a robust model which is able to learn a good policy without knowing the environment [24]. Deep reinforcement learning (DRL) is a very appealing method and methodology in ML, as it works well in dynamic environments such as financial markets (which are extremely dynamic), and can more efficiently identify and learn single-stock trading patterns. Taghian et al. [24] have outlined the following three significant advantages of deep reinforcement learning over other ML techniques:

1.　It does not require previous knowledge of the environment to understand the trade rules;
2.　it can continually adapt to changing environmental scenarios; and
3.　it prioritizes long-term advantages, rather than quick rewards.

The most recent implementations of DRL in financial markets involve continuous or discrete state and action spaces, and utilize one of the following learning methods: critic-only, actor-only, or actor-critic [25]. The critic-only method solves a discrete action space problem, and is the most-utilized learning strategy. The critic-only technique is based on using a Q-value function to learn the optimum action-selection strategy which can maximize the expected future reward, considering the present state. Examples of critic-only methods that focus on training an agent on one stock or asset are deep Q-learning (DQN) and its subsequent modifications [26,27]. However, the critic-only technique has a significant drawback, in that it can only be applied to discrete and finite state and action spaces. This makes it impractical to manage an extensive stock portfolio, as stock prices are continuous [28]. In the actor-only method, however, the agent is the one who immediately discovers and learns the most effective strategy. Rather than learning the Q-value, a neural network is taught the policy. This type of policy is basically a strategy specific to a state based on the probability distribution. In particular, the actor-only method can handle environments with a continuous action space [28].

The goal behind the actor–critic strategy is to concurrently update both the Q-value function and the policy through the actor and critic networks. The critic approximates the

Q-value function, and the actor uses policy gradients to update the probability distribution, as influenced by the critic. Actors become better at making better actions over time, and critics become better at judging those actions in turn. When it comes to trading with an extensive stock portfolio, the actor–critic technique has been shown to be capable of dealing with various complex and challenging trading environments [28].

Wang et al. [7] have formulated a novel deep Q-learning-based strategy, in order to build an end-to-end system to select optimal positions at each trading time step. Using the Deep Deterministic Policy Gradient (DDPG) approach, Xiong et al. [14] have learned a dynamic strategy for stock trading that beat the Dow Jones Industrial Average and min-variance portfolio allocation. Li et al. [18] have investigated the effectiveness of three Deep Q-network variants—including typical DQN, Duelling DQN, and Double DQN—in learning single-stock trading strategies for 10 U.S. equities, and determined that the usual DQN outperformed the other two techniques.

Due to the fact that there are several uncertainties in financial markets, such as changes in the economic policy and misleading corporate information, the direction of the price will be affected by market uncertainty and, so, it is critical to limit the noise coming from the inputs. Deng et al. [16] have proposed a real-time financial signal representation model in an environment which is totally unknown, which uses a recurrent deep neural network to de-noise the price before being trained by RL. Luo et al. [17] have utilized two CNNs to extract features with a DDPG model, in order to learn trading strategies on actual stock-index future data. A novel trading agent developed by Li et al. [29] has been shown to be able to automate the decision processes and set itself up for success in the dynamic financial markets by using stacked de-noise auto-encoders and LSTM (SDAEs-LSTM). They used SDAEs and LSTM as function approximators in order to extract features from high-noise market, resulting in steady and risk-adjusted outcomes in stock and futures markets.

## 3. Methodology and Proposed Model

In this paper, the fundamental concept is modelling time-series data; that is, we consider a sequence modelling problem. In particular, we wish to predict the next closing price, given the previous sequence of (historical) closing prices and other data, by learning a function that describes a temporal dynamics model. At every time step, the model receives one more sequence element as input and, so, this function processes inputs of varying lengths. This function is considered an auto-regressive sequence model, as the outputs of the function depend on its previous outputs. The significant progress in sequence modelling and NLP in recent years—notably with respect to pre-trained Transformer networks such as BERT and GPT-x—has opened up the exciting idea of formalizing sequential decision-making issues in the framework of reinforcement learning (RL). This integration, framing RL as a sequence modelling problem and using a Transformer architecture, provides an effective solution to the problem of long-term credit assignment. According to the long-term credit assignment problem, traditional RL algorithms have difficulty in determining which of the previously executed actions contributed to the return of an episode. However, with the Transformer model, using its attention module allows us to explicitly find relationships between states, actions, and rewards within a sequence, thereby resolving the long-term credit assignment problem. The results presented later in this paper demonstrate the effectiveness of our approach and provide empirical validation.

### 3.1. Preliminaries

3.1.1. Reinforcement Learning

Reinforcement learning (RL) is an area of ML that involves figuring out how software agents should behave in a given environment. This is accomplished by determining what actions should be taken, in order to accumulate the most rewards over a given course of time. Every problem to be solved using RL starts with an environment and one or more agents that can interact with that environment.

The agent first makes observations of the environment, following which it constructs a model of the present state of the environment and the anticipated values of actions that may be taken inside the environment. Then, the agent executes the action determined to have the most significant predicted value (reward), and is rewarded with a sum proportional to the actual worth of the action they choose to do, determined according to the impact that the action has on the environment. The underlying model of the RL agent may then be improved through trial and error (also known as learning by reinforcement), allowing it to eventually learn to perform behaviours that provide a higher level of rewards.

### 3.1.2. The Trading Environment

Typically, a human trader would examine many charts illustrating the price action of a company, sometimes with a few technical indicators superimposed on top (see, e.g., Figure 2). Afterwards, they aggregate and evaluate this visual data to make a well-informed judgment regarding the anticipated movement of the stock.

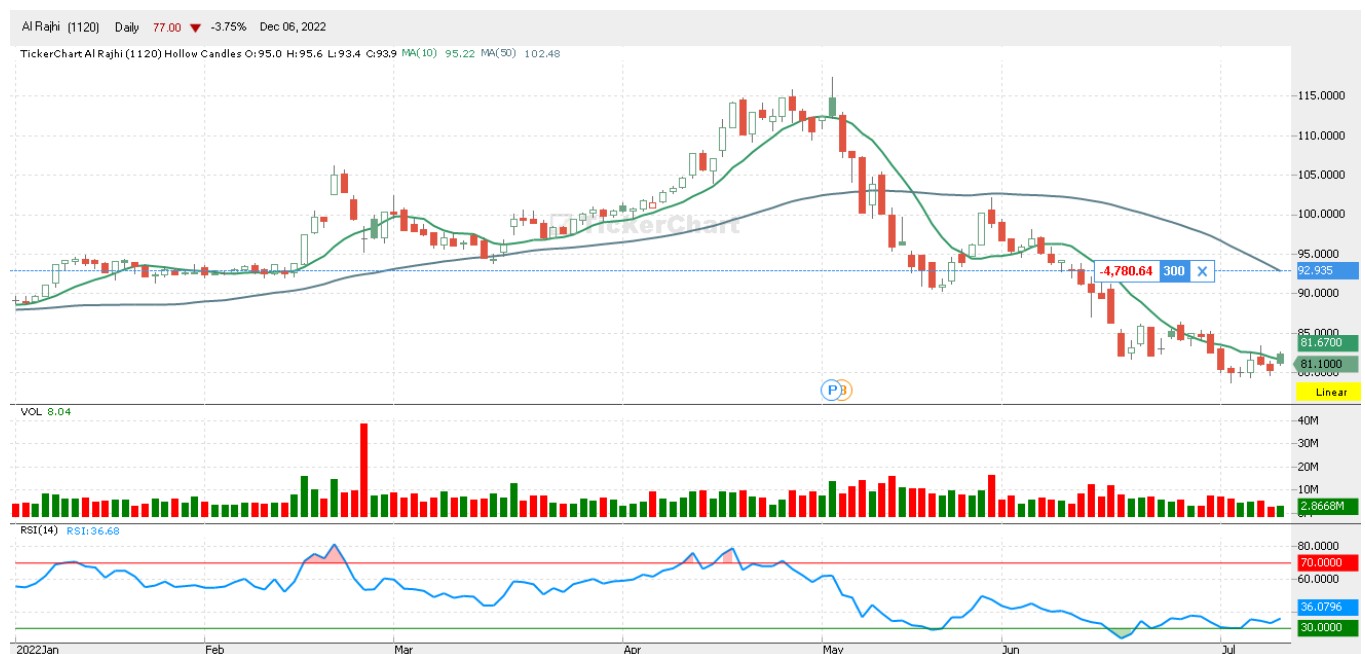

**Figure 2.** Stock Price Chart with technical indicators.

Our trading setting attempts to simulate human behaviour by permitting our agents to monitor the observation or state space, which includes historical daily stock data—Open price, High, Low, Close, and volume (OHLCV)—as well as a few other technical indicators prior to executing (or not executing) a trade. Additional in-depth information on the state of space may be found in Section 3.1.4 below. At each time step, the agent analyses the action of price leading up to the current price and the position of their portfolio, in order to make an informed choice about the next course of action. After the agent has assessed their environment, they must take an action from its action space, including purchase a stock, sell a stock, or hold (do nothing). Section 3.1.5 details the action space in more depth.

### 3.1.3. Reinforcement Learning as Sequence Modelling

The key assumption in this paper is that we will be able to model our environment in the form of a Markov decision process (MDP), specified by $\mathcal{M} = (\mathcal{S}, \mathcal{A}, \mathcal{P}, \mathcal{R})$ where $s \in \mathcal{S}$ represents the state space, $a \in \mathcal{A}$ denotes the action space, $\mathcal{P}$ is the probability distribution over transitions given by $P(s\prime|s, a)$, and $R(s_t, a_t)$ is the reward function. It is important to note that a trajectory $\mathcal{T} = (s_1, a_1, r_1, s_2, a_2, r_2, \cdots, s_t, a_t, r_t)$ in the MDP represents a sequence of states, actions, and rewards that correspond to the agent's prior experience, which are arranged sequentially. Making action predictions from past experience is the

purpose of sequence modelling in RL [30,31]. The probability of reaching state $s\prime$ and earning a reward $r$ following the execution of an action $a$ when beginning in state $s$ is given by:

$$Pr(\hat{a}) = p(a_t | s_{1:t}, a_{1:t}, r_{1:t}). \tag{1}$$

### 3.1.4. State Space $\in \mathbb{R}^{32}$

The representation of the state $\mathfrak{s} = [\mathfrak{f}, \mathfrak{h}, \mathfrak{b}]$ is a vector with 32 dimensions, comprised of the following three components:

- Market Features $\mathfrak{f} \in \mathbb{R}^{28xD}$: Essentially, this is a set of features that is gathered for both tickers and their corresponding market indices, including the transaction date, close price, volume, and six technical indicators. As a result, the feature set includes both the closing price of the ticker and the closing price of the linked index, except for the transaction date, as the ticker and its index will have the same transaction date. Several technical indicators generate multiple values (features) as a result of their calculations, such as Moving Average Convergence Divergence (MACD), which generates two (MCAD) values and a single line. In Section 3.3, we discuss the 28 market features in detail.
- Held shares $\mathfrak{h} \in \mathbb{Z}^{+^{3xD}}$: The total number of shares that are owned in relation to the stocks; this amount (which must be an integer) describes all of the shares possessed.
- Available Balance $\mathfrak{b} \in \mathbb{R}^+$: The amount of liquid assets that are accessible to be used in the process of purchasing or selling a certain stock at each successive time step in the process. This should either be positive or zero, and permitted activities should not result in a balance that is negative.

### 3.1.5. Action Space

There will be a discrete number of action sets within the action space, and these action sets cannot overlap. These action sets are utilized to buy a stock $\mathcal{B}$, sell a stock $\mathcal{S}$, or hold (do nothing) $\mathcal{H}$, where $\mathcal{B} \cup \mathcal{S} \cup \mathcal{H} = \{1, 2, \cdots, D\}$. Furthermore, a continuous spectrum of share quantities can be described as $[-k, 1, 0, 1, \cdots, k]$ where $k$ and $-k$ are the total number of shares that we are able to buy and sell (0–100%, based on the Available Balance $\mathfrak{b}$ and Held shares $\mathfrak{h}$, respectively).

### 3.1.6. Reward Function

It is necessary to provide feedback to the agent through reward signals, in order to teach it which behaviours are appropriate and inappropriate, depending on the context in which they are performed. The agent's trading approach is significantly impacted, both directly and immediately, by this feedback [32]. In our case, the reward function $\mathfrak{r} = (\mathfrak{s}, \mathfrak{a}, \mathfrak{s}')$ the reward that the agent will earn after it has acted in each state. Our goal is to encourage profits that are sustained over a long period of time, while controlling risk as much as possible. The metrics we use to reward our agents are varied, thus promoting our efforts to improve their performance. The measures covered here include the profit and loss function (PnL), the Sharpe ratio, the Sortino ratio, and the Calmar ratio. PnL is a regularly used metric in trading system research to examine the efficiency of reward functions in RL [7,27,33]. Nonetheless, PnL does not consider the risks associated with making a profit. As a result, the risk-adjusted return measure is presumed to be capable of accounting for this factor.

- The Profit and Loss (PnL) The profit function is the one that of the most-used reward functions. Its mathematical formula is as follows:

$$r_t = \left(1 + a_t \times \frac{p_t - p_{t-1}}{p_{t-1}}\right) \frac{p_{t-1}}{p_{t-n}}, \tag{2}$$

where $p_t$ is the closing price on the market at time $t$, while $a_t$ is the agent's action at the same time $t$. Equation (2) is comprises of both one-day and long-term gross returns over $n$ periods.

- Volatility-Based Metrics: Sharpe and Sortino ratio According to [34], the Sharpe ratio is a frequently used measure of the risk-adjusted return, which can indicate both profit and volatility. The Sharpe ratio is calculated by dividing the average risk-free return of the investor by the standard deviation of that return:

$$S_t = \frac{\text{Average} \left( \sum_{i=1}^{W} R_i \right)}{\sigma \left( \sum_{i=1}^{W} R_i \right)}. \tag{3}$$

At time $t$, the Sharpe ratio reward function is $S_t$, the daily return on multiple shares of a stock is $R_i$, and return averages and standard deviations are estimated over returns for periods of $W$, where $W$ denotes the window size, which is used to calculate the average and standard deviation of the returns. Despite this fact, the Sharpe ratio considers volatility in portfolio values. However, the ratio equally regards both upward and downward movements; that is, it also penalizes upward volatility [35–37]. As a matter of fact, upward volatility (upwards price movement) contributes to positive returns, while downward volatility causes losses. In contrast to the Sharpe ratio, the Sortino ratio only considers downward volatility to be a risk, rather than overall volatility. The upward volatility, therefore, is not penalized by this ratio. Mathematically, it is formulated as follows:

$$SR_t = \frac{\text{Average} \left( \sum_{i=1}^{W} R_i \right)}{\sigma_{down} \left( \sum_{i=1}^{W} R_i \right)}, \tag{4}$$

where $SR_t$ and $\sigma_{down}$ represent the reward function of the Sortino ratio at time $t$ and the standard deviation of the daily return below zero for the period $W$, respectively. According to the Sortino ratio, the volatility of the loss under the profit conditions is the only factor considered, as the volatility of the loss under loss conditions is irrelevant. Modern portfolio theory indicates that the Sharpe and Sortino ratios represent high profits without significant volatility. Moreover, according to [38], the Sharpe ratio and the Sortino ratio allow RL agents to perform significantly better over other benchmarks.

However, neither the PnL, Sharpe ratio, or Sortino ratio metrics take into account the maximum drawdown, defined as the maximum loss noted between a peak and a bottom of a portfolio before a new peak is reached. It provides a measure of the rate of change in price over a specified time period, and is an indicator of downside risk [39,40]. A measure known as the Calmar ratio uses maximum drawDown only as a method for quantifying risk [37], where a high Calmar ratio indicates better portfolio performance [41].

### 3.2. Data

In this study, four different industrial indices were paired with individual firms to illustrate the effectiveness of our proposed methodology: The telecommunication services index (TTSI) paired with Saudi Telecom Company, the banking index (TBNI) with Al Rajhi Banking and Investment, the materials index (TMTI) with Saudi Basic Industries Corporation, and the utilities index (TUTI) with Saudi Electricity Company. For all selected indices and companies, the data were obtained by utilizing the Google Finance API for Sheets, covering the same time period of January 2017 through May 2022. Figure 3 depicts how the entire data set was split. We used data from 1 January 2017, to 30 January 2020, for training, and data from 1 February 2020, to 31 December 2020 for parameter tuning and validation. Finally, we tested the performance of our model using trading testing

data, which consisted of unknown out-of-sample data from between 1 January 2021 and 23 May 2022. Our agent continues to be trained as it trades, such that it can adapt to market dynamics better. This is expected to help us to better exploit the trading data.

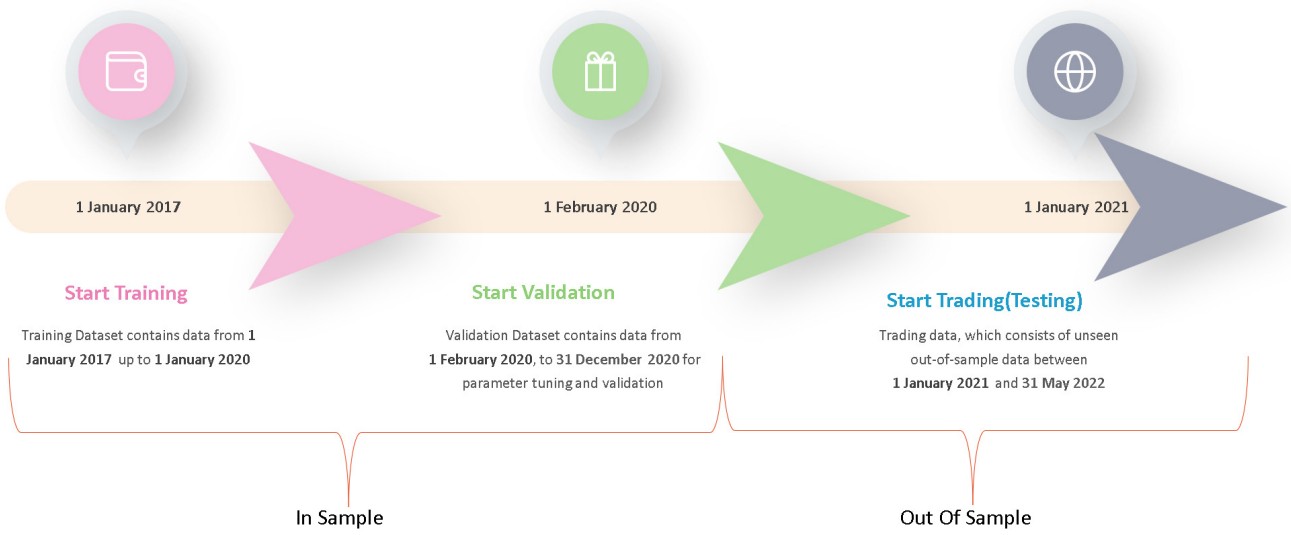

**Figure 3.** Data set split.

In Figure 4, the Tadawul All Share Index (TASI; depicted in blue) is compared to the performance the four selected indices. It is possible to see the changes in each index during the same period. TTSI, TMTI, and TBNI trended either upward or downward in an almost identical direction to TASI. In contrast, for the TUTI index, which reflected movements in the opposite direction, different movements were indicated. The Bank Index (TBNI) began to reduce its gap with the Tadawul All Share Index (TASI) by the beginning of 2019, and surpassed the Tadawul All Share Index by the end of 2021, mainly due to the growth of the Saudi economy. In addition to being one of the 20 largest economies in the world, Saudi Arabia is actively participating in the G20, due to its economic growth. The robust financial system, the effective banking system, and the vast number of government companies run by highly qualified Saudis have made Saudi Arabia one of the world's most prominent players in the global economy and oil markets.

The next step was to manually examine the stock performances during the training and validation period. Figures 5–8 show the change in performance of each stock over the course of the training and testing periods. The green vertical line indicates the start of the validation phase, while the red vertical line indicates the beginning of the testing phase (data not seen by the model).

Figure 5 for Saudi Telecom Company shows an upward trend beginning in 2017 and continuing until the middle of 2019, at which point it begins to show a declining trend that continued until the first quarter of 2020. In contrast, for Saudi Electricity Company, Figure 6 shows a downward trend until the end of 2018, followed by drifting sideways until the middle of 2020 and reversion to an upward trend beyond that point.

Furthermore, Figure 7 illustrates that the value of Al-Rajhi Banking and Investment stock was maintained (upward trends), even during the COVID-19 pandemic, as a result of strong financial standings. Over the past three years, the bank has issued bonus issues twice. The most recent capital increase was in February 2012, when SAR 15 billion in retained earnings were injected, and five shares were issued for every five held [42].

On the other hand, the stock price of the Saudi Basic Industries Corporation (Figure 8) plunged from 130.40SR in May 2018 to 66.10SR in March 2020. Following a slow start in March 2020, the stock entered April 2020 on a relatively high note, which gave way to the recent downward trend. After beginning 2021 in the 104SR range, the stock price

increased throughout the year and hit its current all-time high when it surpassed 134SR on 21 October 2021.

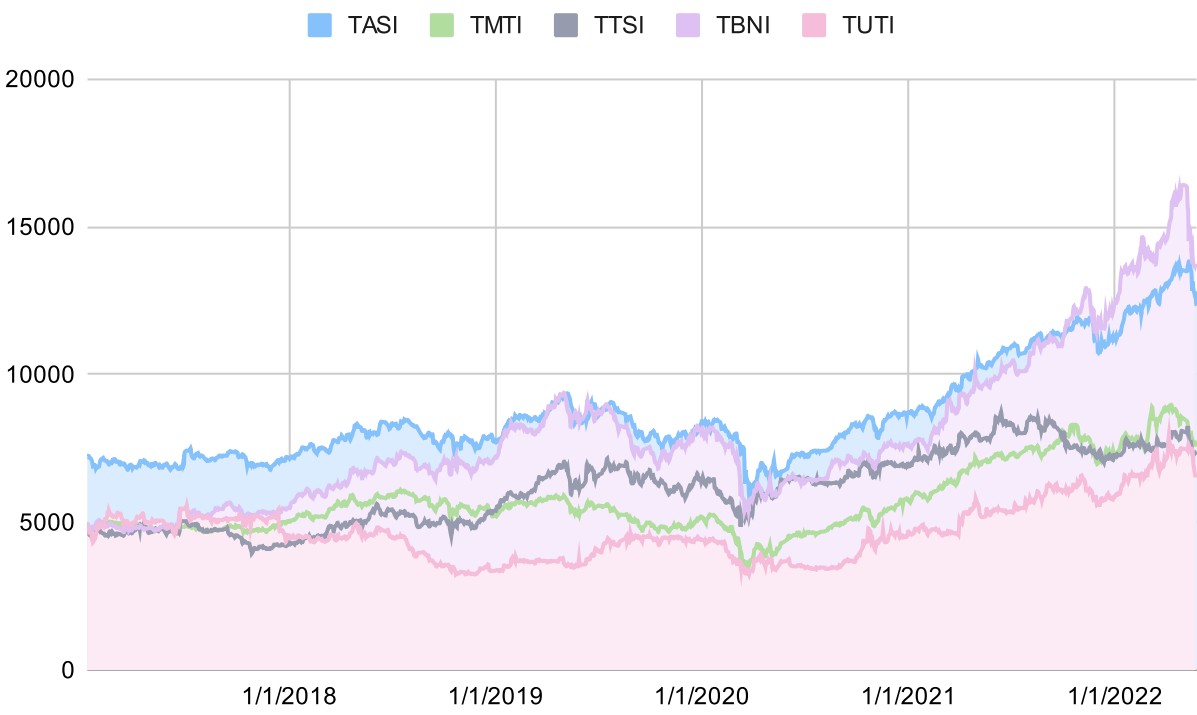

**Figure 4.** Comparison of performance of the four selected indices with that of the Tadawul All Share Index (TASI).

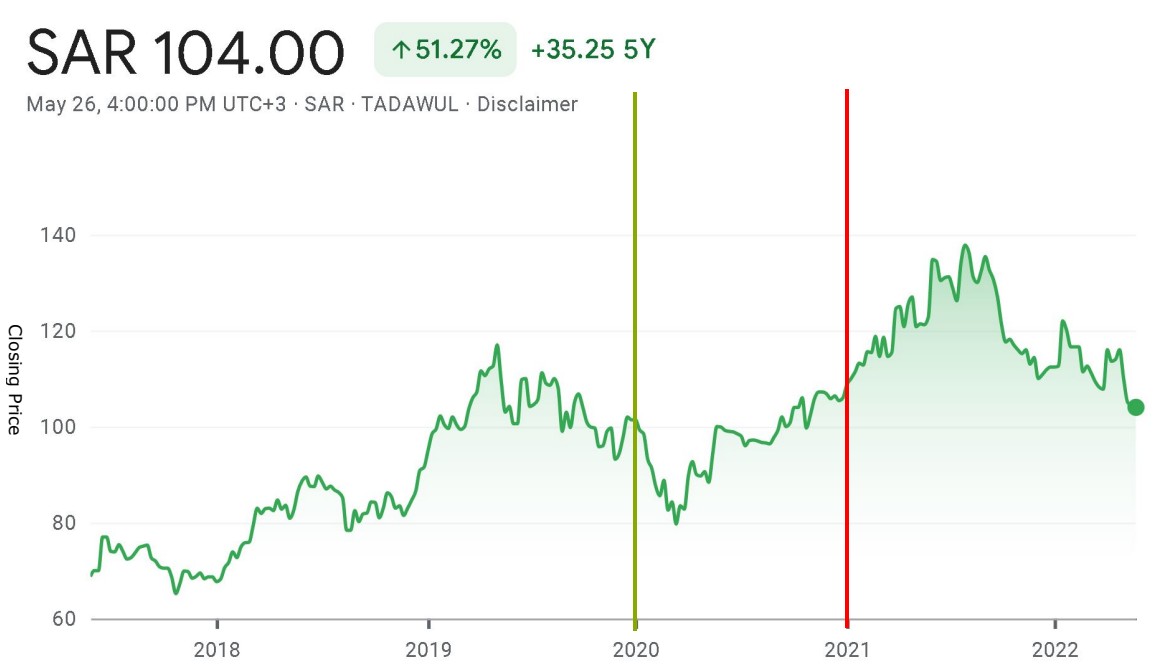

**Figure 5.** Behaviour of Saudi Telecom Company (7010) during the training and testing periods.

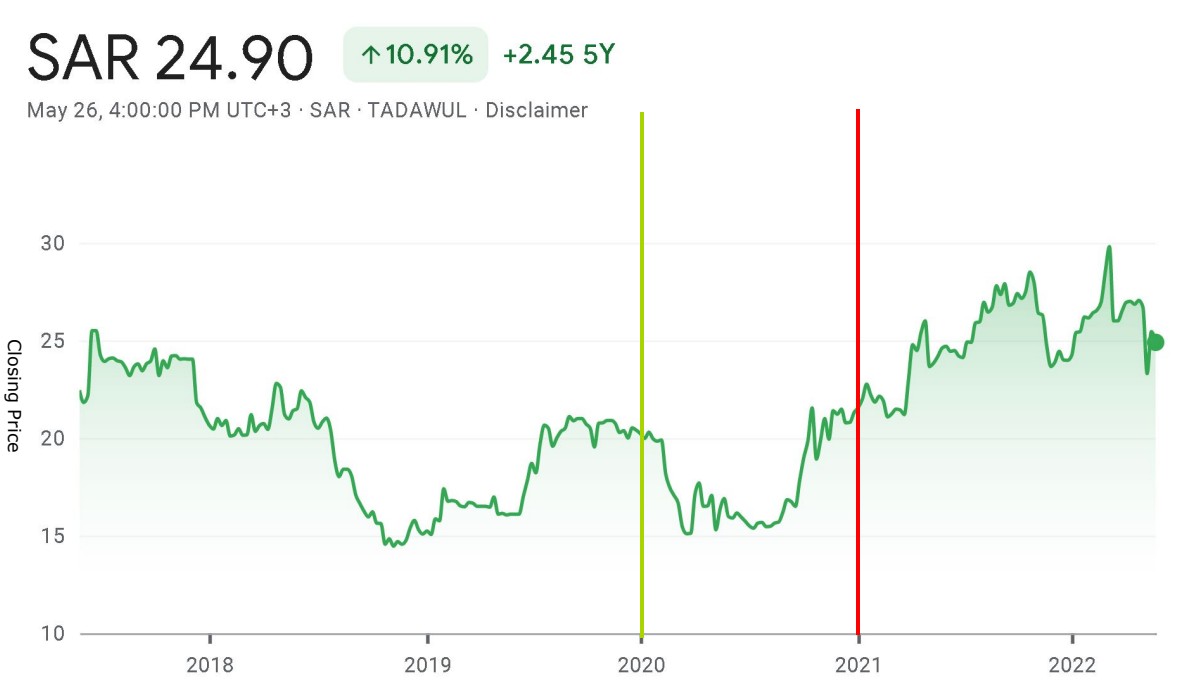

**Figure 6.** Behaviour of Saudi Electricity Company (5110) during the training and testing periods.

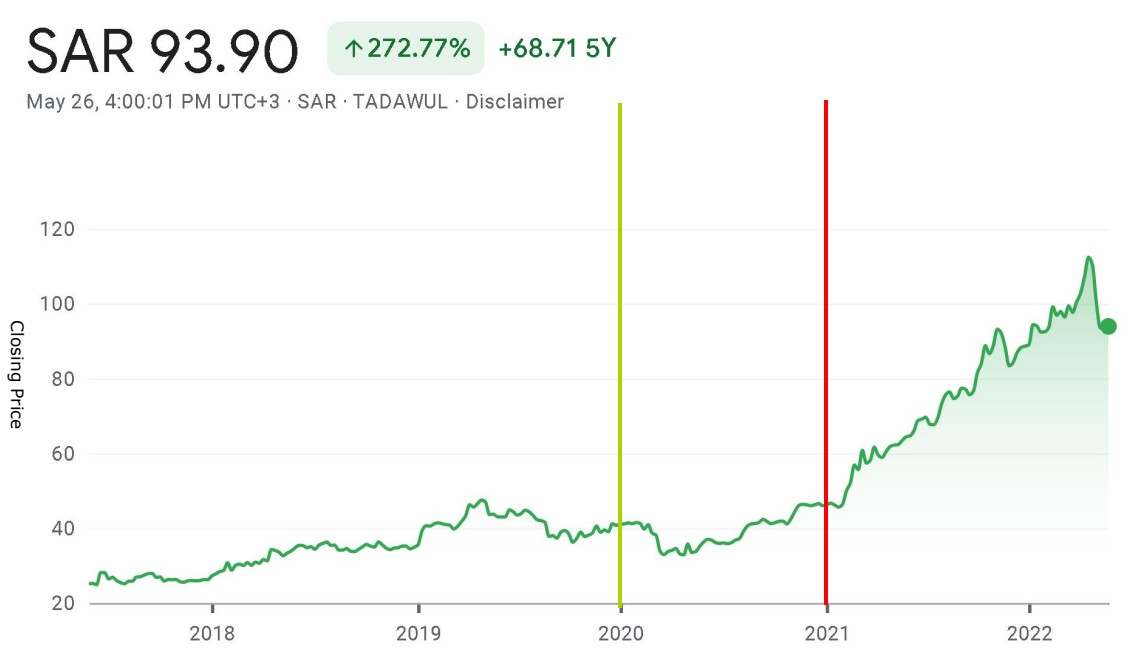

**Figure 7.** Behaviour of Al-Rajhi Banking and Investment (1120) during the training and testing periods.

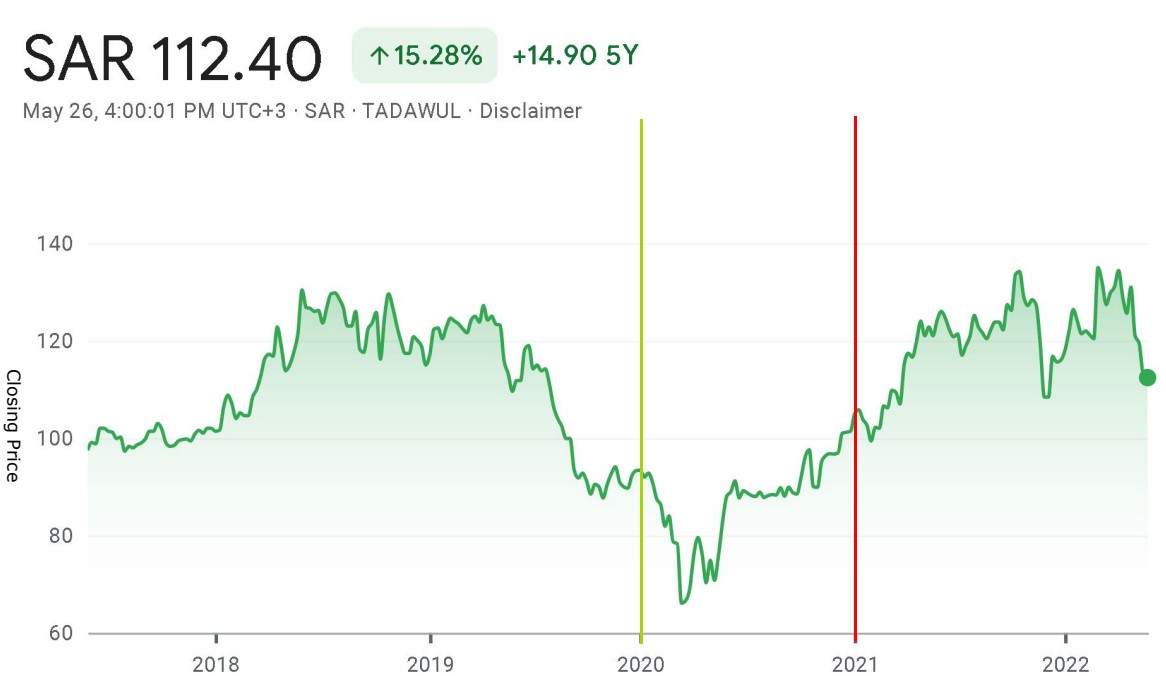

**Figure 8.** Behaviour of the Saudi Basic Industries Corporation (1020) during the training and testing periods.

### 3.2.1. Data Pre-Processing

According to [43], adjusting the data pre-processing to accommodate ML algorithms can significantly contribute to the performance of such algorithms, having a fundamental impact on their performance. It has been noted, however, that despite its critical importance, this work is perceived as one of the most routine and laborious of all ML activities [43,44]. There are several methods for pre-processing data, including Data Imputation, Data Normalization, and Feature Extraction. In the following, we describe the various pre-processing steps.

### 3.2.2. Data Imputation

In many modern ML applications, understanding how to deal with incomplete data sets is one of the most prominent issues. Missing data may be caused by various factors, such as flawed acquisition method, worries about data privacy, lack of knowledge, non-availability of information, and so on. This creates inconsistencies within the data set, hence influencing the predictions. Silva-Ramirez et al. [45] have defined imputation as the process of substituting estimated values for missing data points in a data set. Imputation can create a complete and consistent data set through the reconstruction of missing data. According to [46], missing data may be broken down into one of the three following categories:

1. Missing Completely at Random (MCAR): The missing data are not related to the values of any other variable in the data set.
2. Missing at random (MAR): The probability of the missing values of a variable is dependent on another variable in the data set, but not on that variable.
3. Missing Not at Random (MNAR): That missing values in a variable are closely related to the variable and not the other variables in the data set. This is the most concerning missing value.

In our situation, there may be gaps in the data set due to missing stock data or technical indicators. In this case, data imputation becomes critical for a comprehensive and consistent data set, which will substantially influence prediction. In addition, it is a well-known fact that the Tadawul stock markets are closed on the weekends (Friday and Saturday), on national holidays, and on other days that are considered public holidays. This results in a scenario in which price data pertaining to stocks are unavailable on specific days when the market is closed. The loss of these data may result in loss of information. The handling of missing data can be accomplished in several ways. However, we used linear interpolation in all instances that did not have available stock data in our situation, which was done to acquire an accurate data representation. The process of creating new data points from existing ones is referred to as linear interpolation. This approach is one of the simplest missing data methods available, as it only requires two known data points to calculate the missing value.

### 3.2.3. Data Stationarity

According to [47], real-world data such as stock prices provide some of the most complicated and challenging problems in forecasting. Real-world data are inherently non-stationary, which means that their distributions change over time, making precise forecasting difficult. The data distribution may be utterly unpredictable due to non-stationary data characteristics such as trends, seasonality, residual, random walks, and their combinations. This is problematic for ML, as it goes against one of the foundational assumptions: The training, validation, and test data sets are all drawn from the same distribution, and every sample has an equally distributed distribution [48].

Looking more closely at Figures 4–8, it is evident that all of our indices and stocks exhibited a trend. In fact, Augmented Dickey–Fuller is a frequently used method to determine whether a time series is stationary or not [49]. Prior to carrying out the preceding step, we used the statistics library module (statmodels.tsa.seasonal) to examine the characteristics of the stock data. The time-series was divided into three components by this module, as depicted in Figures 9–12, for each ticker: trend, seasonality, and residuals. Trends define the overall direction of a series throughout time. Seasonality refers to patterns that are repeated at regular intervals according to seasonal elements (e.g., annual, monthly, or weekly). After eliminating the preceding components, the residual indicates the irregular component of the time-series.

The bottom line is that our stock data contained noticeable trends and seasonality, impacting the ability of our algorithm to predict the stock data accurately. This conclusion was verified by the Augmented Dickey–Fuller test, a sort of statistical test known as a unit root test. This test reveals how firmly a trend characterizes a time-series. The objective of a statistical test is to assess whether or not a null hypothesis can be rejected, and the null hypothesis for this test is that a unit root cannot represent the stock data and that it is not completely stationary (i.e., it has some structure which is time-dependent). Alternatively, the stock data may be stationary, thus rejecting the null hypothesis. Interpreting this result is guided by the *p*-value obtained from the test. Whenever the *p*-value is less than a threshold, we can reject the null hypothesis, demonstrating that the data are stationary. Meanwhile, a *p*-value greater than the threshold indicates that the null hypothesis cannot be rejected (non-stationary).

For the aim of confirming the stationarity of the stock data, the Augmented Dickey–Fuller unit root test was carried out, and the outcomes of this test are shown in Table 2.

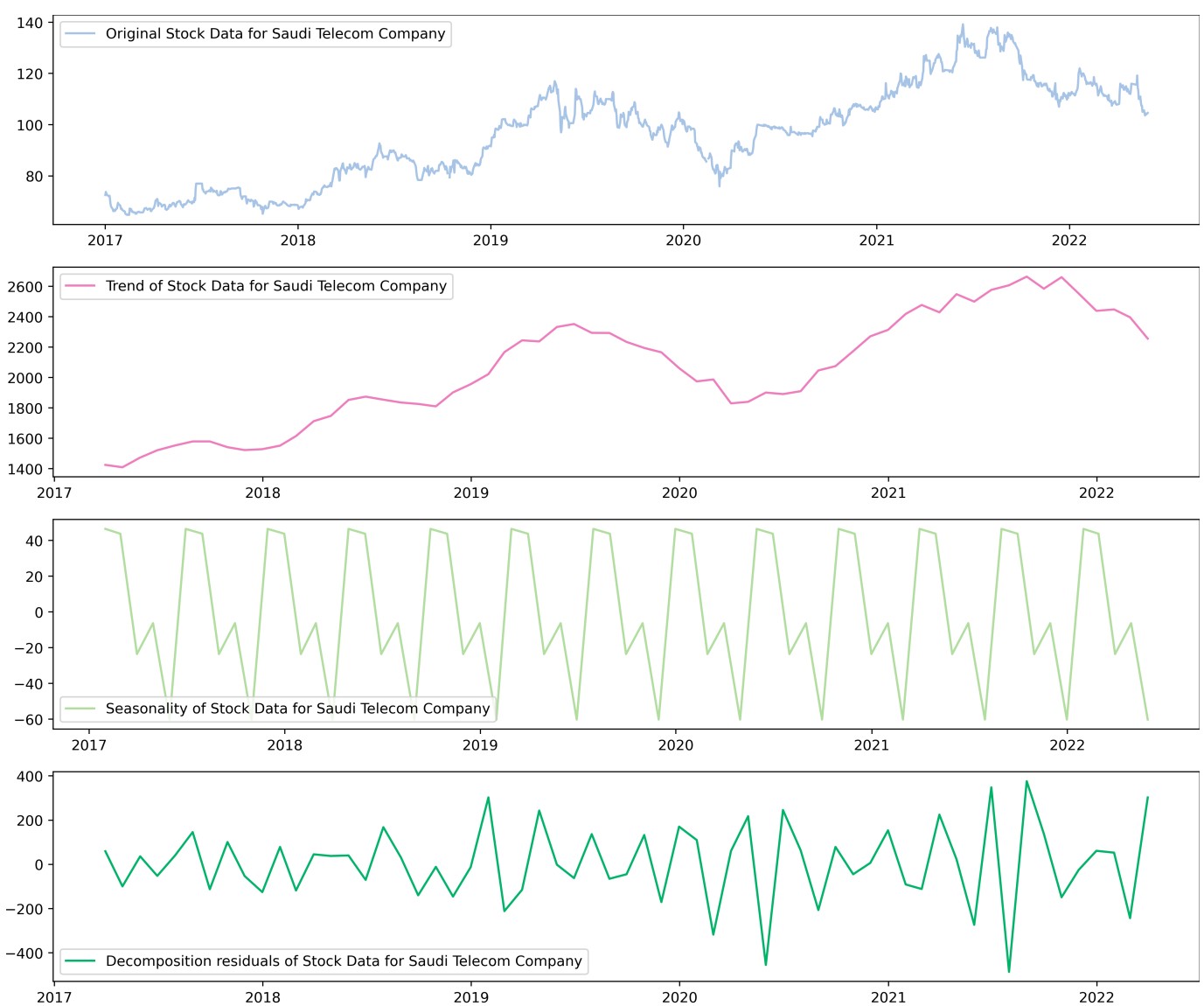

**Figure 9.** Decomposition of 7010 Stock Data: Trend, Seasonal (Periodic), and Residual Components.

**Table 2.** Results of Augmented Dickey–Fuller (ADF) Unit Root Test.

| Index/Stocks | ADF Statistic | *p*-Value | Critical Values | Decision |
|---|---|---|---|---|
| Telecommunication Services Index(TTSI) | −1.161312 | 0.690013 | 1%: −3.435 5%: −2.864 10%: −2.568 | Failed to Reject $H_o$, Time-Series is Non-Stationary |
| Saudi Telecom Company (7010) | −1.594104 | 0.486555 | 1%: −3.435 5%: −2.864 10%: −2.568 | Failed to Reject $H_o$, Time-Series is Non-Stationary |
| Banks Index (TBNI) | 1.187172 | 0.995896 | 1%: −3.435 5%: −2.864 10%: −2.568 | Failed to Reject $H_o$, Time-Series is Non-Stationary |
| Al Rajhi Banking and Investment Corp(1120) | 2.972744 | 1.000000 | 1%: −3.435 5%: −2.864 10%: −2.568 | Failed to Reject $H_o$, Time-Series is Non-Stationary |
| Materials Index (TMTI) | −0.562991 | 0.879127 | 1%: −3.435 5%: −2.864 10%: −2.568 | Failed to Reject $H_o$, Time-Series is Non-Stationary |
| Saudi Basic Industries Corporation (2010) | −1.878952 | 0.342027 | 1%: −3.435 5%: −2.864 10%: −2.568 | Failed to Reject $H_o$, Time-Series is Non-Stationary |
| Utilities Index (TUTI) | 0.334678 | 0.978881 | 1%: −3.435 5%: −2.864 10%: −2.568 | Failed to Reject $H_o$, Time-Series is Non-Stationary |
| Saudi Electricity Company (5110) | −1.661178 | 0.451217 | 1%: −3.435 5%: −2.864 10%: −2.568 | Failed to Reject $H_o$, Time-Series is Non-Stationary |

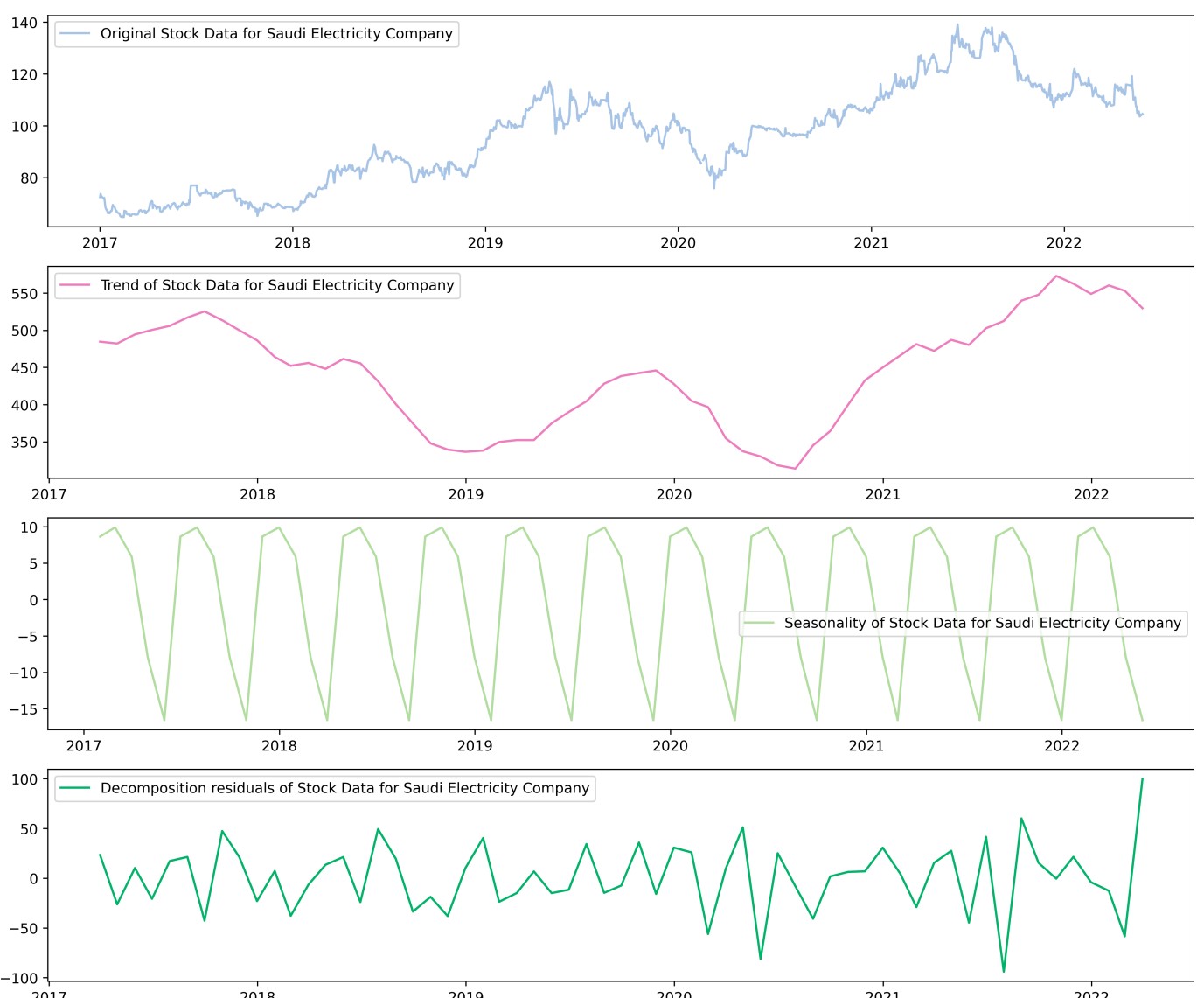

**Figure 10.** Decomposition of 5110 Stock Data: Trend, Seasonal (Periodic), and Residual Components.

Table 2 makes it quite clear that the null hypothesis was not rejected for the data of indices and stocks, confirming that our data were not stationary. This can be observed by the fact that the absolute value of the ADF-statistic was less that the absolute value of the ADF-critic at the 1% level (3.592462), as well as by the fact that the *p*-values were higher than 0.05. Data that are non-stationary can be handled using various techniques, including differencing and transformation. The purpose of a difference is to eliminate variance in the mean, where subtracting the current observation ($x_t$) from the prior observation ($x_{t-1}$) yields the difference. The mathematical expression for differencing is as follows:

$$x_t = x_t - x_{t-1}. \tag{5}$$

Transformations, however, are used to stabilize non-constant variances by applying some mathematical function to each time-series value to remove a pattern. The most common transformation methods are log transforms, square roots, and power transforms, including the Box–Cox transformation and the Yeo–Johnson transformation. In this study,

we employ Yeo–Johnson transformations. Mathematically speaking, the Yeo–Johnson transformation may be expressed as:

$$y_i^{(\lambda)} = \begin{cases} \left( (y_i + 1)^{\lambda} - 1 \right) / \lambda & \text{if } \lambda \neq 0, y \geq 0 \\ \log(y_i + 1) & \text{if } \lambda = 0, y \geq 0 \\ -\left[ (-y_i + 1)^{(2-\lambda)} - 1 \right] / (2 - \lambda) & \text{if } \lambda \neq 2, y < 0 \\ -\log(-y_i + 1) & \text{if } \lambda = 2, y < 0, \end{cases} \tag{6}$$

where $y_i$ denotes the observed feature transformed using a parameter $\lambda$. Maximum likelihood is typically used to estimate $\lambda$ in the Yeo–Johnson transformation, assuming that the transformed variable follows a normal distribution [50]. Nonetheless, according to [51], the application of a transform on its own does not necessarily cause the data to become stationary. This conclusion was reached using the ADF test. Therefore, to make data stationary, differencing was also applied in combination with the transform.

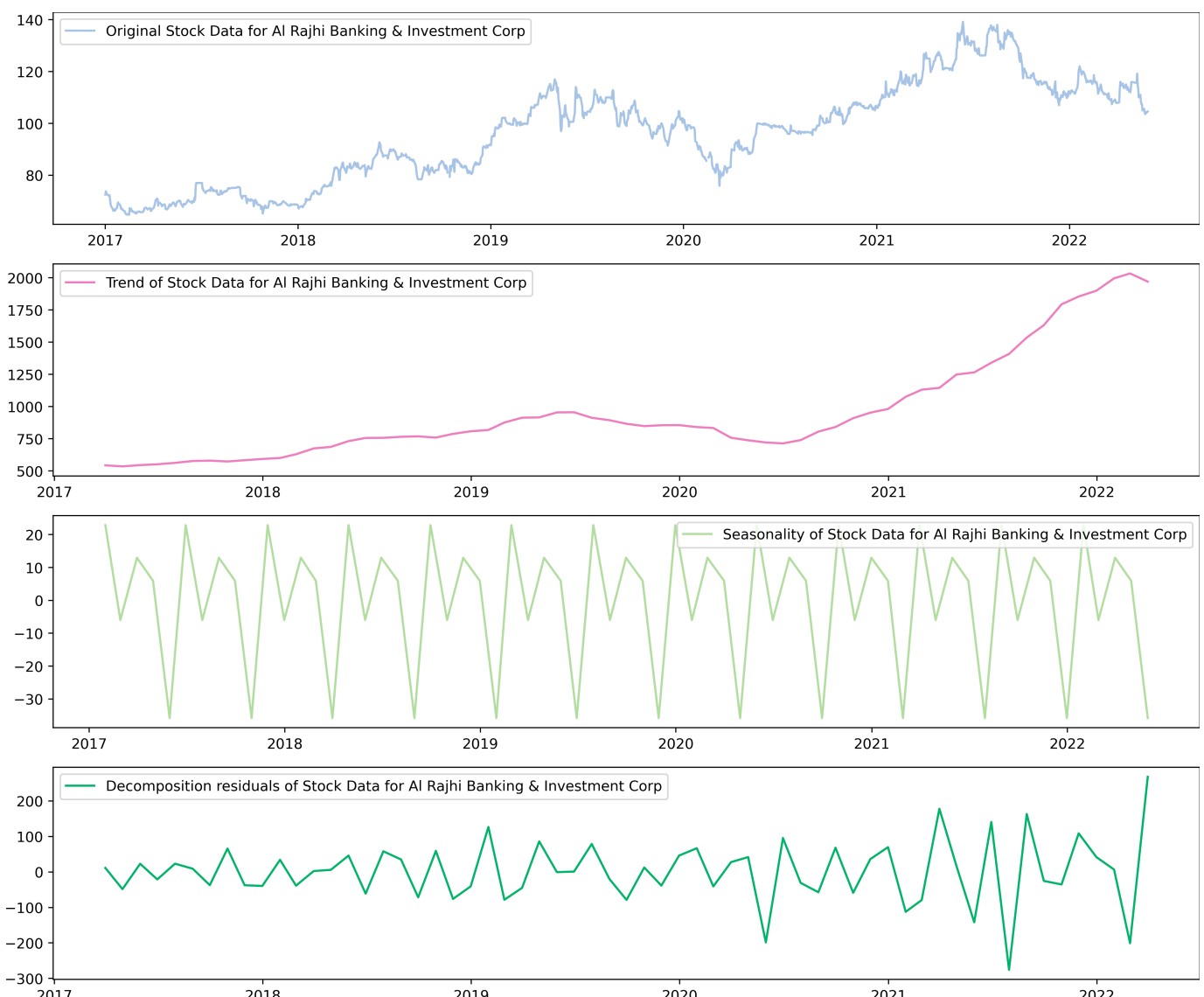

**Figure 11.** Decomposition of 1120 Stock Data: Trend, Seasonal (Periodic), and Residual Components.

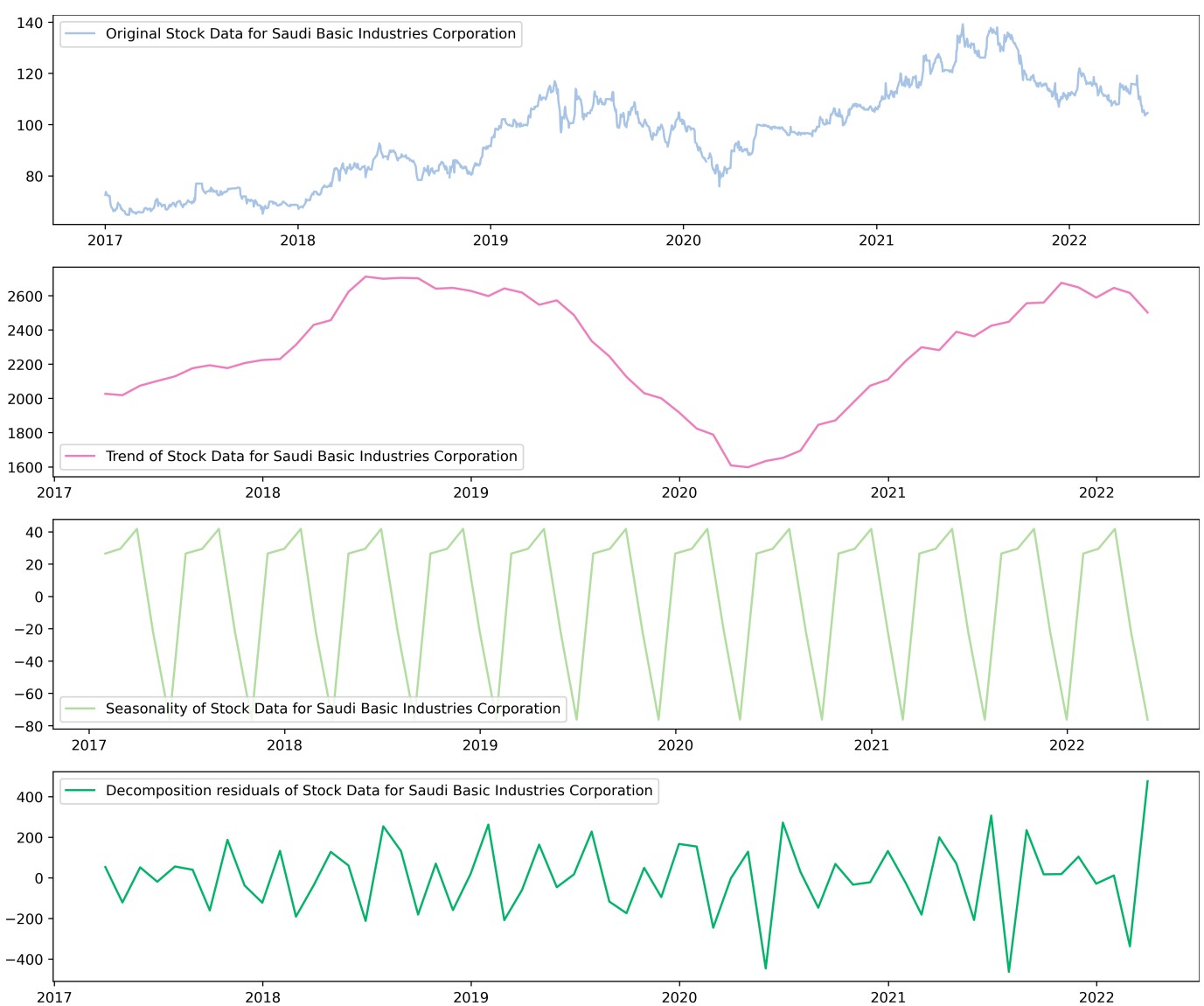

**Figure 12.** Decomposition of 2010 Stock Data: Trend, Seasonal (Periodic), and Residual Components.

### 3.2.4. Data Normalization

Data normalization is the process which involves changing the range of values in a data set. When we utilize price and volume data, all stock data should fall within a standard value range [12]. Generally, ML algorithms converge more quickly or perform better when the data they are working with are close to being normally distributed and/or when the scales on which they are working are the same. The activation function in an ML algorithm, such as the sigmoid function, also has a saturation point, at which the outputs become constant [12,52]. To use model cells effectively, one must normalize the inputs before applying them. This operation was completed using the MinMaxScaler method included in the scikit-learn package. When the MinMaxScaler algorithm is applied to a feature, it takes the minimum value and subtracts it from each value in the feature. Then, it divides the range by the result of this subtraction. Therefore, the range of a feature is the difference between the highest possible value and the lowest possible value. The structure of the initial distribution is kept intact through the use of this method by MinMaxScaler. In particular, the input values were normalized by the MinMaxScaler such that they fall in the range $[0, 1]$.

*3.3. Feature Extraction*

This subsection focuses on describing how features are extracted from the stock data, which will be used later to perform the analysis. We start by outlining the selection of our sets of features. In [45], feature extraction was defined as the process of obtaining derived values (features) from the input data and producing unique and informative properties to enhance the learning and generalization tasks of ML algorithms. Stock data are generally subject to a high amount of noise, and we can reduce the effect of noise by using appropriate technical indicators. These technical indicators can also provide a different perspective on the market, based on their use, and provide a higher degree of predictive ability to predict future price movements. Statistically, these characteristics should generally have a reasonable correlation with market movements, which will complement our forecasted data. Combining these features should lead to a nice mix of practical observations, from which our model can benefit.

Inputs to a Transformer network are used to more effectively capture trends in the stock market. These inputs include the daily closing price, trading volume, and six technical indicators for each ticker and its linked index. These technical indicators have been proposed by [53]. It is usual practice to classify technical indicators into four categories: Volume, Volatility, Momentum, and Trend. Summaries of the technical indicators that were utilized are given below:

1.  Exponential moving average (EMA) These are utilized to reduce the noise and point out the short- and long-term trends in time-series data. The exponential moving average EMA $(x, \alpha)$ is generated by exponentially reducing the weight of observations $X_i$ regarding their distance from $X_t$ using a weighted multiplier $\alpha$.

$$\text{EMA}(x_t, \alpha) = \alpha x_t + (1 - \alpha) \text{EMA}(x_{t-1}, \alpha).$$
$$\alpha = \frac{2}{N+1} \tag{7}$$

2.  Money flow index (MFI): Based on price and volume, the money flow index determines the amount of money moving into and out of a specific ticker, or, to put it another way, if a given stock has been over-bought or over-sold. This is what is known as a momentum indicator. When the MFI is over 80, it indicates an over-bought condition; meanwhile, when it is below 20, it suggests an over-sold condition. The MFI may be computed using the following formula:

$$\text{MFI} = 100 - \frac{100}{1 + \text{MFR}},$$
$$\text{where MFR} = \frac{\text{Positive Money Flow}}{\text{Negative Money Flow}} \tag{8}$$
$$\text{Money Flow} = \left( \frac{\text{High} + \text{Low} + \text{Close}}{3} \right) Volume.$$

3.  Relative strength index (RSI): This index is also used as a momentum indicator, which determines whether a ticker is over-bought or over-sold by considering both the velocity and magnitude of price fluctuations. Its value may vary anywhere from 0 to 100, with low values indicating a stock that is being over-sold and high values indicating an over-bought stock. The following formula may be used to easily determine the value of this indicator:

$$\text{RSI} = 100 - \frac{100}{1 + \text{RS}},$$
$$\text{where RS} = \frac{\text{Average of Up closes}}{\text{Average of Down closes}}. \tag{9}$$

4.　Moving average convergence-divergence (MACD): The MACD is another indicator used to illustrate the relationship between two exponential moving averages (EMAs): Slow ($\theta_1$) and fast ($\theta_2$). According to [54], the moving averages are calculated based on the following criteria: $\theta_1$ comprises 26 periods (market standard), $\theta_2$ consists of 12 periods (usual for the financial markets), $\theta_1 - \theta_2$ for constructing the MACD line. Finally, the moving average is constructed using the MACD line (standardized with 9 periods). This indicator can be used to assess the trend-following momentum within a stock. The formula for calculating this indicator is:

$$\text{MACD} = \text{EMA}(\theta_1) - \text{EMA}(\theta_2). \tag{10}$$

5.　Commodity channel index (CCI): The commodity channel index (also abbreviated as CCI) is a type of trend indicator that calculates the difference between the average of historical prices and the current price value. When the CCI is greater than zero, the current price is higher than the average value of historical prices; conversely, when the CCI is lower than zero, the price is lower than the historical average. A reading above 100 is above the buy threshold, and that below $- 100$ is below the sell threshold.

$$\text{CCI} = \frac{1}{0.015} \frac{P_{typical\,price} - \text{SMA}(P_{typical\,price})}{\text{MAD}(P_{typical\,price})}, \tag{11}$$

where SMA is the simple moving average and MAD is the mean absolute deviation.

6.　Ichimoku: The Ichimoku Hinko Hyo indicator identifies the trend direction and determines accurate support and resistance levels. There are five main components of the Ichimoku Cloud indicator that provide reliable trade signals: Kijun-Sen, Senkou Span B, Senkou Span A, Tenkan-Sen, and Chiou Span.

A Python library for technical analysis [55] was used to calculate these technical indicators based on raw stock data, comprised of opening price, the closing price, the low price, the high price, and the trading volume.

*3.4. Decision Transformer Model*

Traditional deep reinforcement learning agents are trained to optimize decisions to achieve the optimal return. At every time step, an agent observes the environment and decides what action to take to help itself achieve a higher return magnitude in future interactions. For this study, we trained an RL agent using Decision Transformers [30] as our base model. A decision transformer is a type of sequence model that forecasts future behaviour by taking into account both Return-To-Go (RTG) and previous interactions between an agent and its environment.

A Decision Transformer produces optimal actions by mapping diverse experiences to their respective return magnitudes during training, as opposed to conventional RL, which computes policy gradients or fits value functions. Using a variety of experiences when training an agent increases the model's exposure to a wide range of trading variations, thereby helping it to derive useful trading rules that will enable it to succeed under any given circumstance. In this case, based on the desired return (reward), past states, and actions, the Decision Transformer will be able to generate any return value in the range it has observed during training, including the optimal return. By using this Decision Transformer model, we can generate future decisions based on the desired return (reward), past states, and actions.

We summarise the model architecture in Figure 13, where returns, states, and actions are incorporated into modality-specific linear embeddings, and a positional episodic time step encoding is also added. The model processes a trajectory, $\mathcal{T} = (s_1, a_1, r_1, s_2, a_2, r_2, ....s_t, a_t, r_t)$, in order to learn meaningful patterns. Rather than directly feeding rewards, the models are fed with the sum of future rewards, resulting in trajectory representations which can be trained and generated by autoregressive methods. The return of the trajectory determines the initial Return $r_1$. During training, at time step $t$,

the model uses the tokens from the last transformer context length ($K$), which is a hyperparameter, to predict $a_t$ using a cross-entropy loss function, and the average of the losses is computed at each time step. Throughout the evaluation, we specify the initial state $s_1$ and the target return $r_1$. The model then generates the action $a_t$. Upon executing the generated action, the target reward is subtracted from the achieved reward to determine the next state. This process continues until the episode ends.

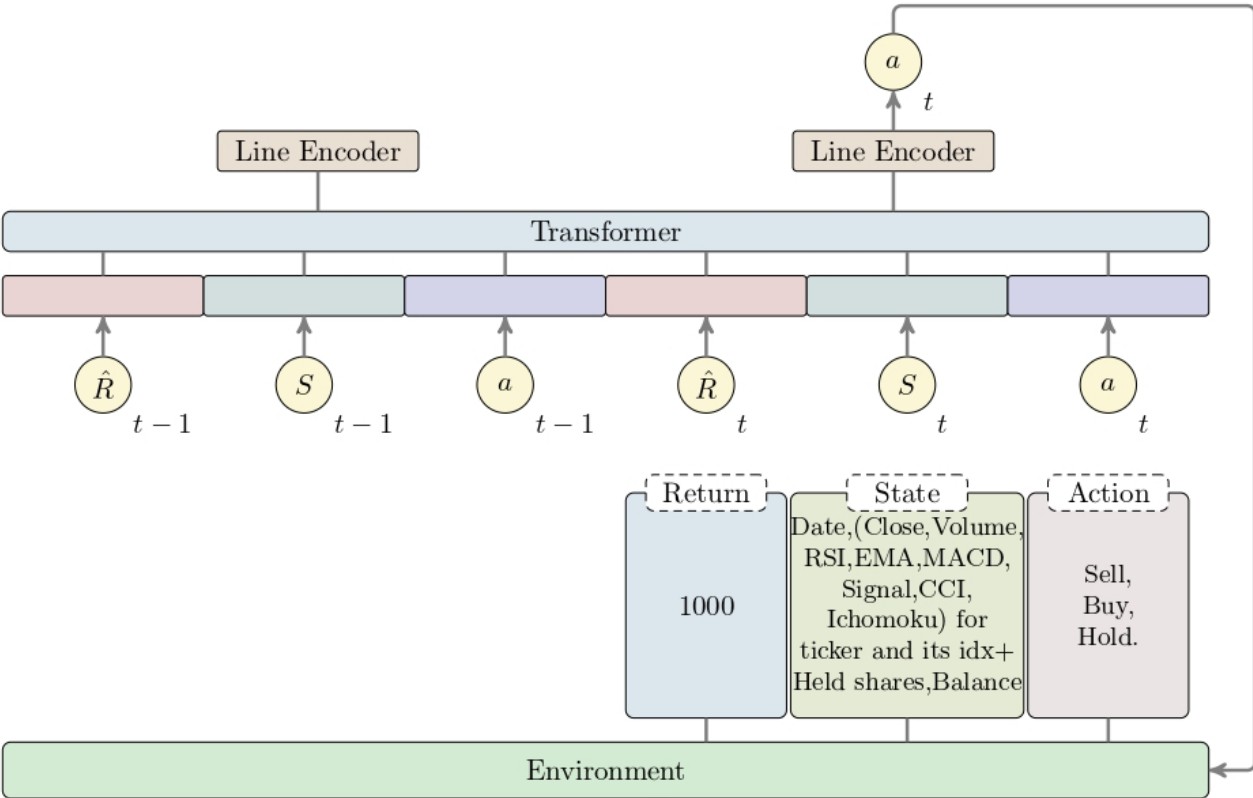

**Figure 13.** Model Architecture.

## 4. Experiments and Analysis

In this section, we describe the setup utilized for our experiments, present the numerical results of our model on the selected stocks, and test the algorithm against different spread settings to investigate the usefulness of the model. The objective of this section is to validate our approach with respect to the Saudi stock market. In our opinion, our approach can be applied to other financial markets with little modification.

### 4.1. Hyperparameters

A transformer's performance is affected by various parameters, including its dimensionality, the activation functions, the number of embeddings, hidden states, learning rate, and the number of attention heads within the attention layer. These parameters are referred to as hyperparameters. Optimizing hyperparameters can be achieved through different techniques, including grid-search, Bayesian, random, and genetic algorithms, among others [56,57]. The high computational costs precluded us from performing a systematic hyperparameter tuning. Parameter pruning is a common method for reducing the model search and hyperparameter optimization times [56]. As part of our experiment, we cloned the same hyperparameters as the Decision Transformer [30], in order to quickly find an initial parameter set. In Table 3, all hyperparameter values and descriptions are provided.

**Table 3.** Values of various hyperparameters used in our Model.

| Hyperparameter | Value |
| --- | --- |
| State dimensionality | The state size for the DRL environment |
| Action dimensionality | The size of the action space = 3 |
| Number of hidden layers | 12 |
| Number of attention heads | 12 |
| Learning Rate | 0.001 |
| Optimizer | GELU |
| Batch size | 256 |
| Dropout probability | 0.1 |
| Layer normalization epsilon | $1 \times 10^{-5}$ |

Additionally, the hyperparameters and model architecture remained constant throughout the experiments.

*4.2. Model Training*

We discuss model training in this section. The model was trained using a fixed data set $D$ including the previously gathered trajectory data. From the data set, the Decision Transform extracted $n$ mini-batches of length $K$ for each stock and created token embeddings before normalizing them through the normalization layers. Tokens were also embedded with the embedded time step at each step. GPT was then used to process the tokens. A forward pass was applied to each mini-batch in the architecture, which predicts the action (i.e., whether to sell, buy, or hold). Accordingly, the loss between the prediction and the subsequent actual action in the data set was calculated as Cross-Entropy (CE) loss.

We used seven different reward functions during model training. The technique proposed in this paper is able to automatically choose the best-performing reward function on the basis of the model performance. Based on the selection of reward functions, model hyperparameters optimization is carried out. We used the Sortino ratio, omega, the Calmar ratio, normal, max drawdown, cumulative returns, and annual volatility as reward functions, where the cumulative returns indicate the total return at the end of the trading phase; annual volatility and maximum drawdown are indicators of a model's robustness; the Sortino ratio is a popular statistic that considers both return and risk; the Calmar ratio is a risk-quantification measure calculated based on the max drawdown; and the normal reward function is the reward without risk adjustment, calculated as the current net worth—the net worth after the ticker is sold or bought.

The performance of the model under the outcome of net worth is illustrated in Figures 14–17 for each stock, based on the Sortino ratio, the Calmar ratio, omega, and annual volatility reward functions. In the case of the Saudi Telecom Company (Figure 14), the Sortino ratio demonstrated the best reward function, with a maximum net worth exceeding 16% of the initial balance of $10,000 and an average over 17%. For Saudi Electricity Company (Figure 15), the performance under the Sortino ratio and annual volatility was the same. Both showed a maximum net worth of slightly over 35%. However, in the last few months, the Sortino ratio outperformed the annual volatility. Saudi Basic Industries Corporation (Figure 17) showed a maximum net worth of slightly over 12% of the initial account balance for the annual volatility reward function. However, Al-Rajihi Banking and Investments showed the highest annual volatility ratio, with a net worth of more than 13%.

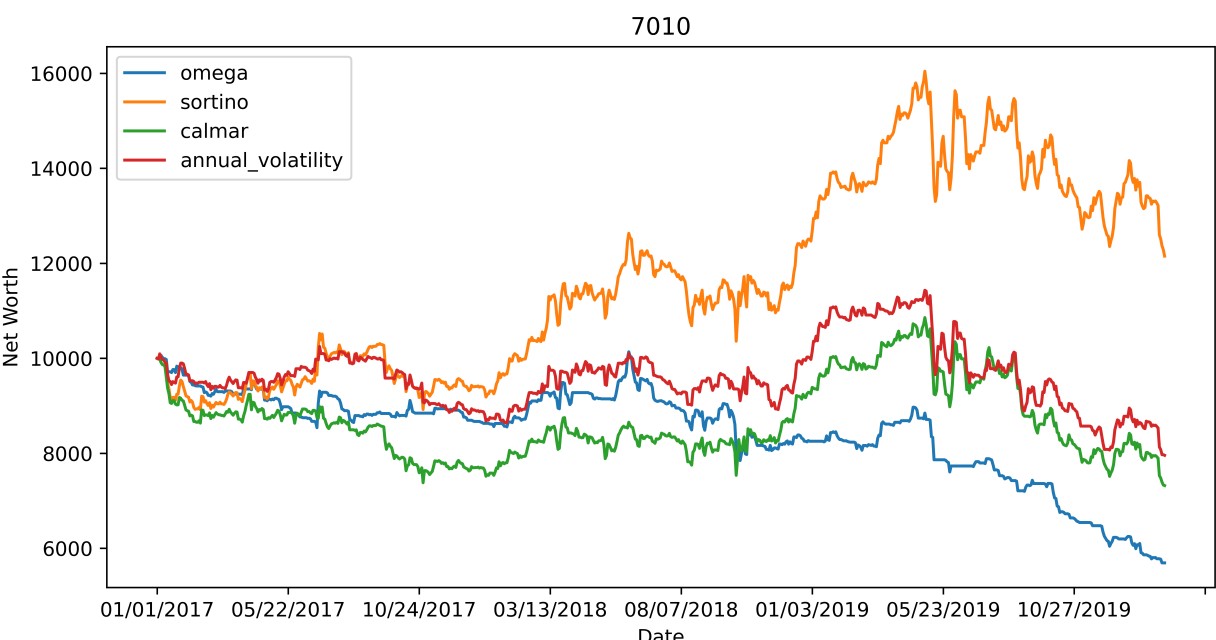

**Figure 14.** Outcome of 7010 stock net worth during model training with different reward functions (Initial portfolio value $10,000).

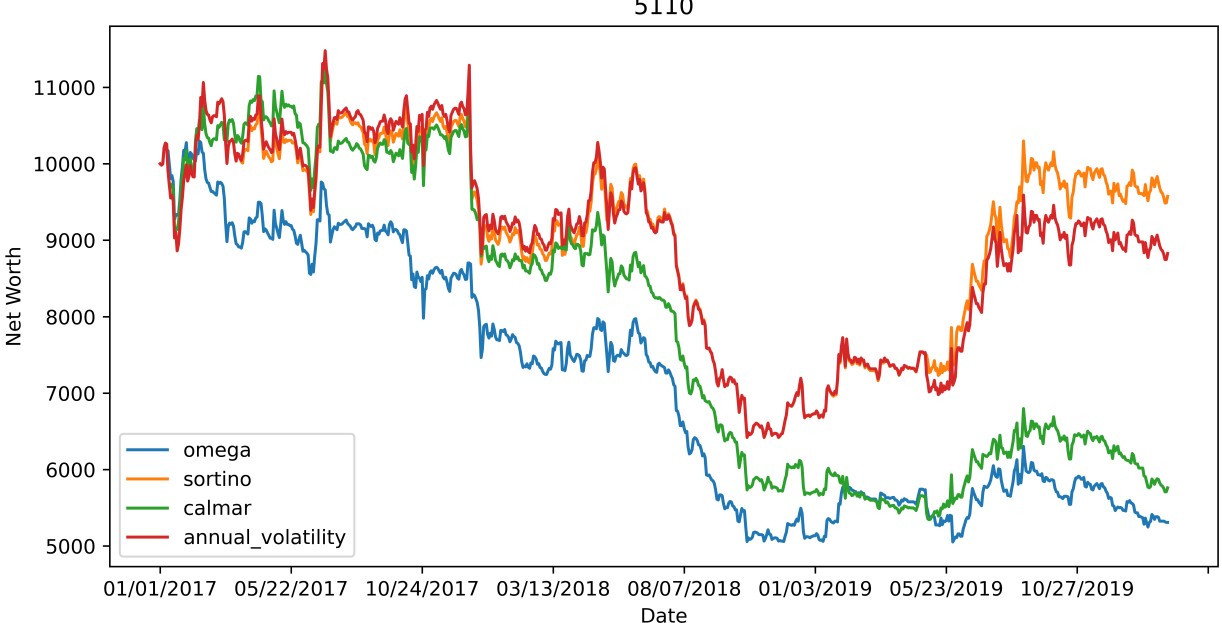

**Figure 15.** Outcome of 5110 stock net worth during model training with different reward functions (Initial portfolio value $10,000).

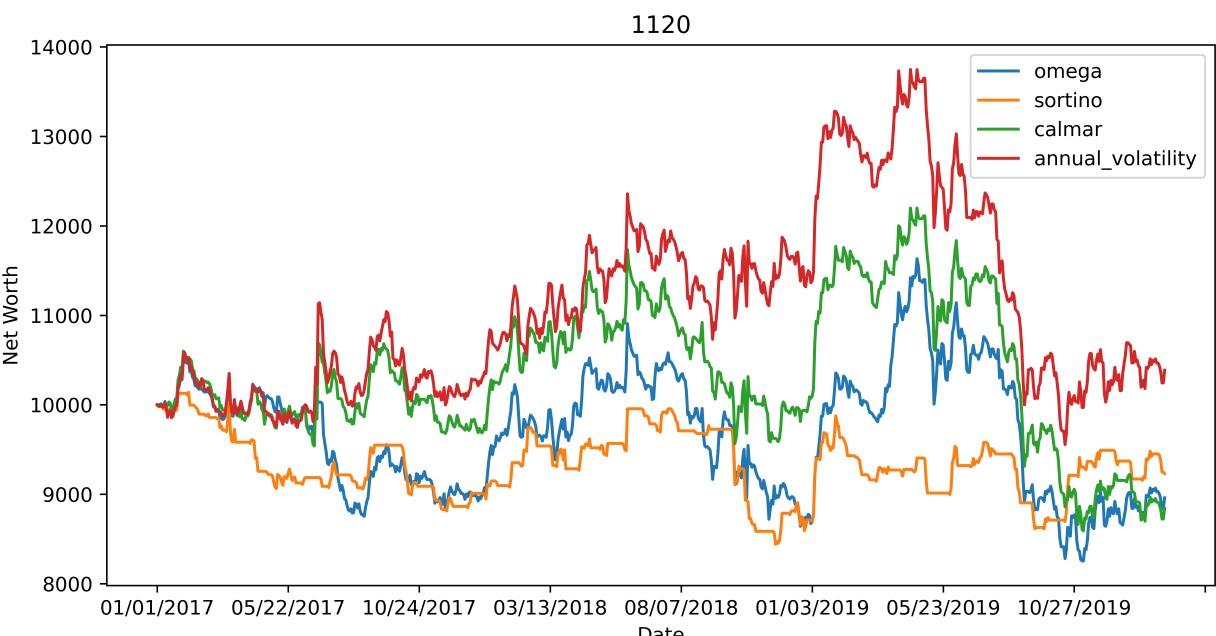

**Figure 16.** Outcome of 1120 stock net worth during model training with different reward functions (Initial portfolio value $10,000).

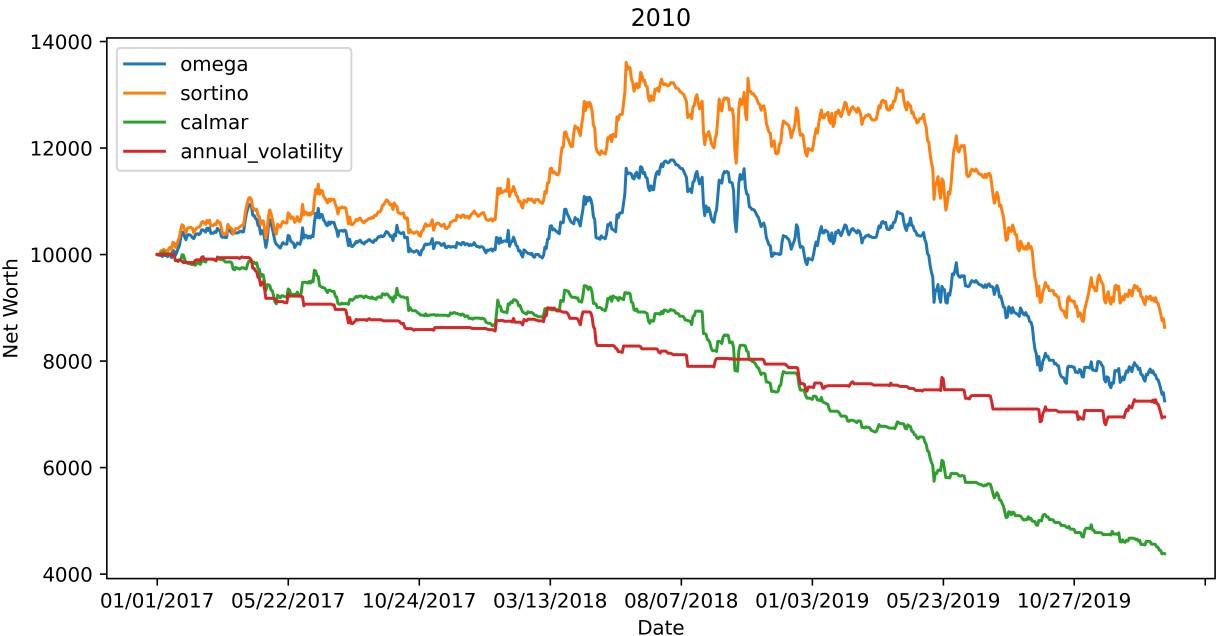

**Figure 17.** Outcome of 2010 stock net worth during model training with different reward functions (Initial portfolio value $10,000).

Figures 18–21 illustrate the outcome of target return during model training. As the goal of model was to maximize the profit over a long period of time, each reward function attempted to increase the return on the initial value of a portfolio (i.e., $10,000 for each company). For Saudi Telecom Company (Figure 18), at the start of training (around the samples of one year), all reward functions performed equally. From the middle of year 2018, the Sortino ratio and cumulative return increased the value of return by 12% of the initial value. However, in the end, the Sortino ratio demonstrated the best reward

function, with a maximum net worth exceeding 16% of the initial value. In the case of Saudi Electricity Company (Figure 19), annual volatility and normal reward function showed equal performance, with a return value slightly over 11% of the initial investment. For the stocks of Al-Rajihi Banking and Investment (Figure 20), cumulative returns showed the highest return value, exceeding 18% of the initial account balance. In the case of Saudi Basic Industries Corporation (Figure 21), the highest return (of more than 14%) was observed for the Sortino ratio reward function.

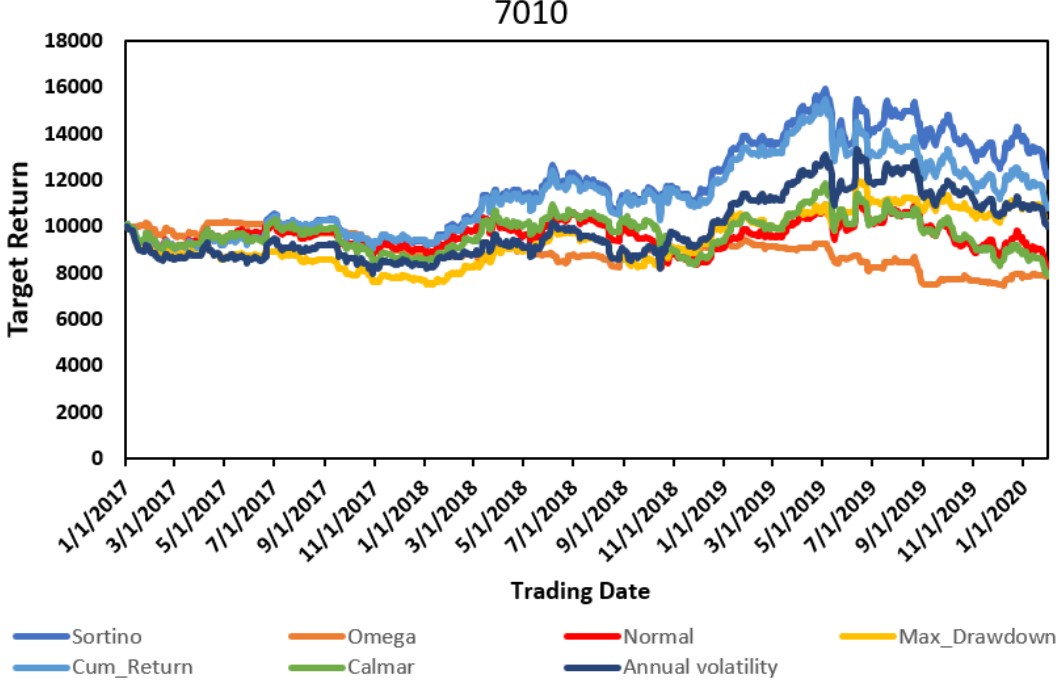

**Figure 18.** Outcome of 7010 stock target return during model training with different reward functions (Initial portfolio value $10,000).

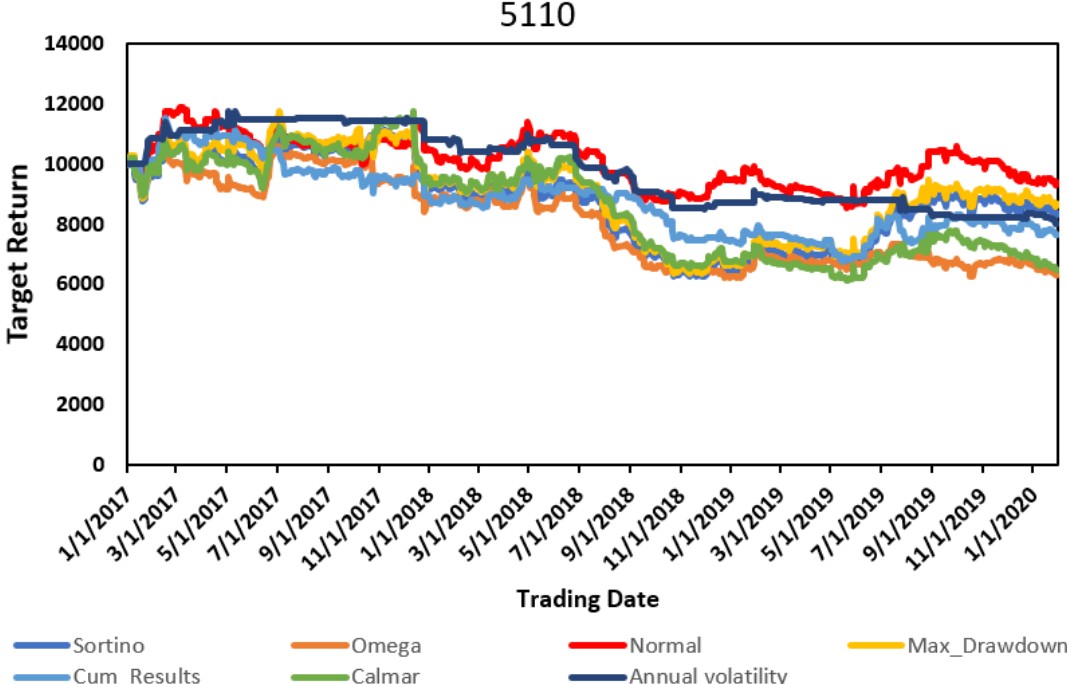

**Figure 19.** Outcome of 5110 stock target return during model training with different reward functions (Initial portfolio value $10,000).

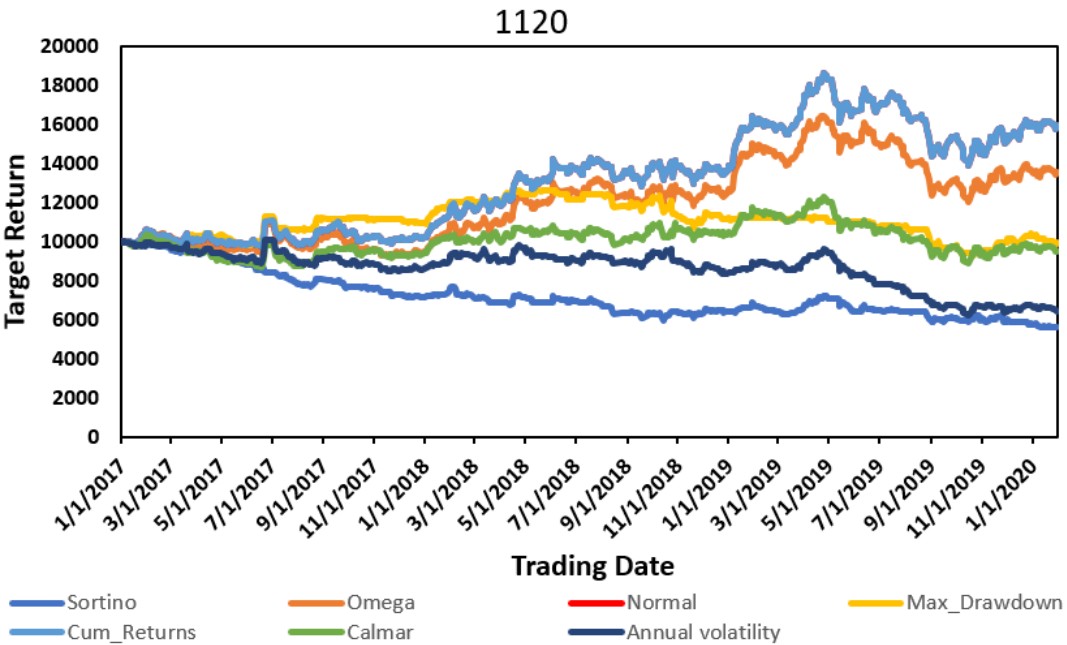

**Figure 20.** Outcome of 1120 stock target return during model training with different reward functions (Initial portfolio value $ 10,000).

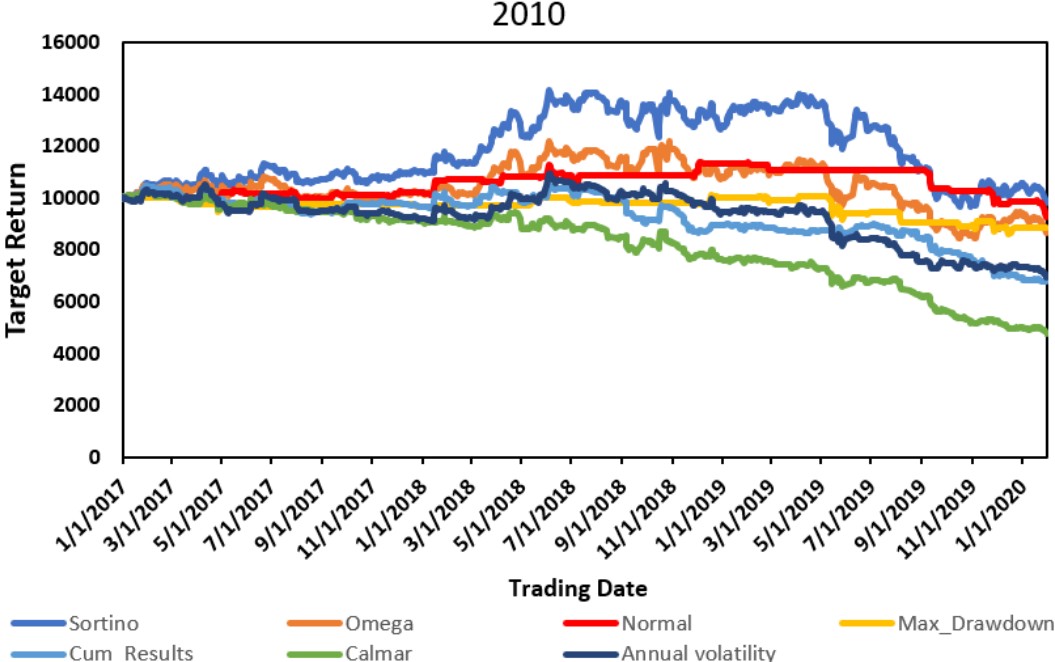

**Figure 21.** Outcome of 2010 stock target return during model training with different reward functions (Initial portfolio value $ 10,000).

### 4.3. Complexity Analysis

The Decision Transformer was quadratically scaled by the sequence length (L) factor required for atom functioning of the transformer attention technique—namely, the canonical dot-product—which led the time complexity and memory consumption per layer to be $O(L^2)$ [58,59]. Under certain circumstances, this is an issue that can be critical to achieving the goal of solving experimental tasks under certain computational settings, as the complexity may lead to significantly longer average response times.

## 5. Discussion

In this section, we discuss the performance of our model on the out-of-sample test data set, the limitations of our work, and future research directions. During the training of the model, our technique automatically selected the best reward function to increase the net worth and profit on the initial investment. This automatic selection of the best reward function allowed the model to learn the optimal hyperparameters. According to our discussion in the previous section, the Sortino ratio, max drawdown, annual volatility, and cumulative returns showed the best performance for both outcomes (i.e., net worth and target return). All of these reward functions ensure the robustness of the model, along with risk management in the return. The performance associated to all the reward functions, along with their descriptions, are summarized in Table 4. The average increase in the net worth of each stock is also provided in the table. The Sortino ratio showed the highest average increase in the case of all stocks, with an average increase of 21.54% for Saudi Telecom Company, 18.54% for Saudi Electricity Company, 17% for Saudi Basic Industries Corporation, and 19.36% for Al-Rajihi Banking and Investment.

Note that, in the reinforcement network, failures are represented by agents not achieving the target return. These failures were analysed using the following return functions: Sortino ratio, max drawdown, annual volatility, and cumulative return. All of these reward functions ensure the robustness of the model, in addition to risk management on the return.

**Table 4.** Reward functions used in this work, in order of their performance (best to worst from top to bottom).

| Reward Function | Description | Average Increase in Net worth | | | |
|---|---|---|---|---|---|
| | | 7010 | 5110 | 2010 | 1120 |
| Sortino ratio | Measures the risk on return by penalizing downside volatility | 21.54% | 18.54% | 17% | 9.36% |
| Cumulative returns | Indicates the total return at the end of the trading phase. | 8.02% | 10.61% | 12.92% | 14.57% |
| Annual volatility | Indicator of model robustness and shows the annual standard deviation of portfolio return. | 7.84% | 10.61% | 12.92% | 14.57% |
| Calmar ratio | Risk quantification measure. High ratio indicates better portfolio performance. | 1.02% | 5.81% | 10.90% | 4.75% |
| Omega | Weighted gain to loss probability ratio at a specific value of expected return. | 8.12% | 0.61% | 2.92% | 1.57% |
| Max Drawdown | Weighted gain to loss probability ratio at a specific value of expected return. | 0.024 | 1.61% | 1.28% | 0.07% |
| Normal | Reward without risk adjustment. | 10.02% | 1.68% | 15.46% | 2.67% |

Figures 22–24 show the results for net worth on the testing data samples. Saudi Telecom Company (Figure 22), Saudi Electricity Company (Figure 23), and Saudi Basic Industries Corporation (Figure 24) showed maximum increases of 13%, more than 14%, and slightly over 13% of the initial net worth, respectively. These results indicate that our model can effectively predict the trading of stocks to maximize profit in the long run, as the net worth follows an increasing trend for almost 10 months in each case. Notably, even with a slight decrease in the value for a few months, the trend of net worth began increasing again. Hence, our model is effective for trading suggestions to provide profit for long-term investments by balancing risk.

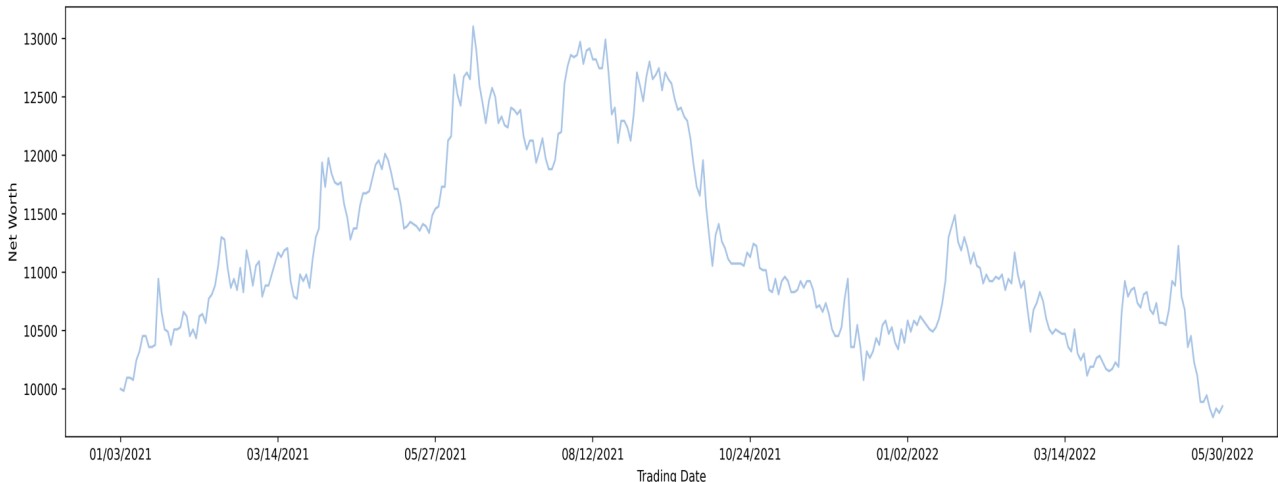

**Figure 22.** 7010 stock net worth prediction on test data set (Initial portfolio value $10,000).

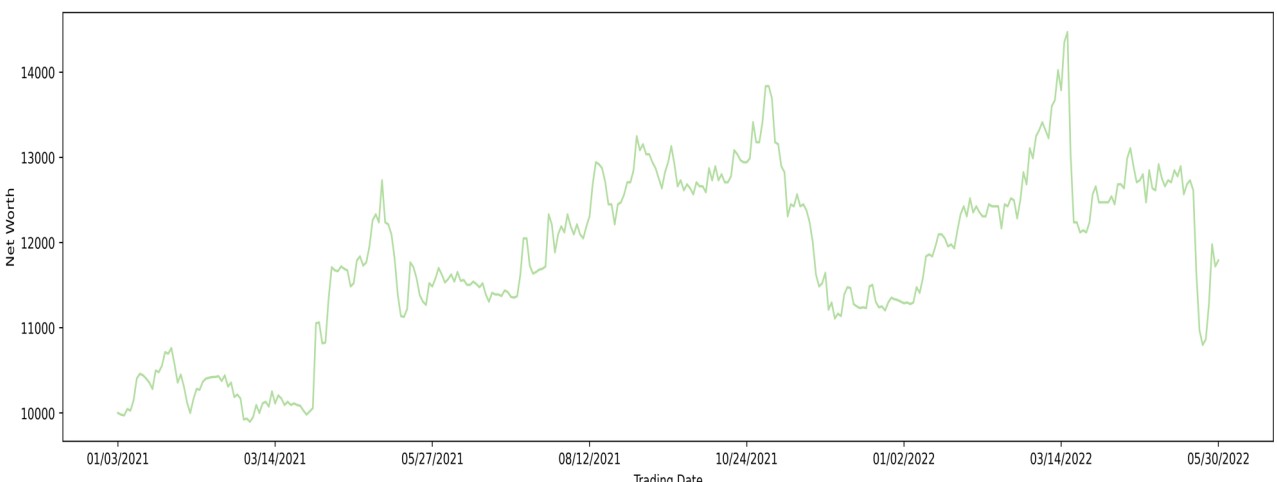

**Figure 23.** 5110 stock net worth prediction on test data set (Initial portfolio value $10,000).

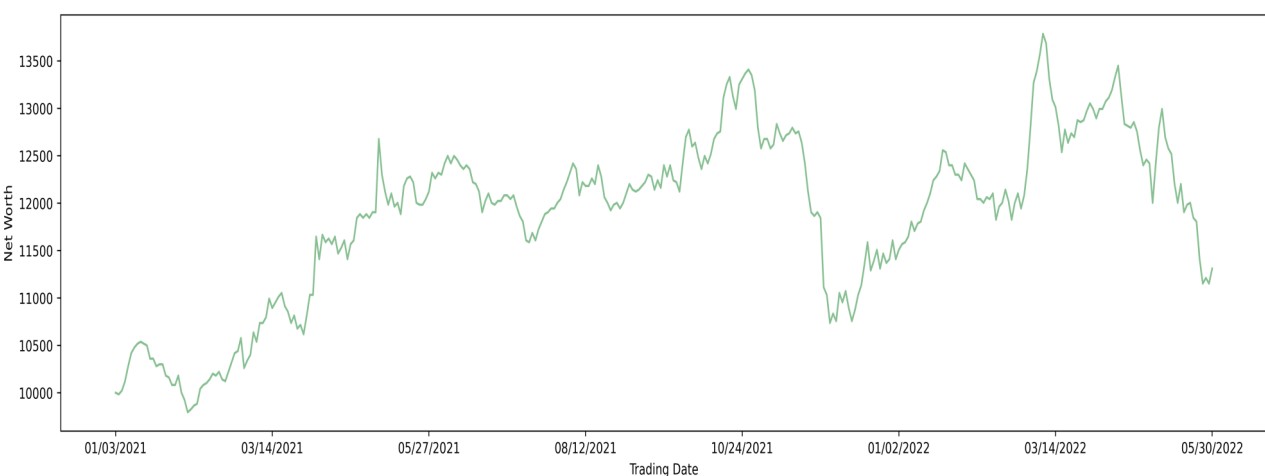

**Figure 24.** 2010 stock net worth prediction on test data set (initial portfolio value $10,000).

It is evident from the results that our algorithm is capable of generating consistent profit, considering all the transactions. Moreover, among the reward functions, the Sortino

ratio, cumulative returns, and annual volatility were the best, based on their performance and parameter optimization. One can use the proposed technique for the prediction of stock trading for other companies, using these reward functions, or the model can be further optimized, according to their data.

In the future, we intend to work on strengthening our model by enhancing it to detect times of crisis, such that it can make decisions which are risk-averse under crisis situations. We will also implement and test our technique for other trading strategies, such as break-even and trailing stop, using data from other stocks. We also hope to predict the best trading time using our technique for bullish and bearish trading markets.

*Comparison with Related Works*

Many significant breakthroughs at the frontier of machine learning have relied on large and diverse data sets and benchmarks, such as the General Language Understanding Evaluation (GLUE) benchmark, which provides a suite of tools to evaluate the performance of NLP models across a variety of modern NLU tasks. However, the potential of enormous and diverse data sets and benchmarks in reinforcement learning (RL) is yet to be established [60,61]. The current practice is to collect RL data sets and create custom environments that are task-specific. As a result, the generalization capabilities of RL models are challenged [62,63]. In a nutshell, three variables that influence the learnt policy's generalization potential have been identified [62]: Training set size, neural network size, and regularization. Therefore, we compare our work to others by focusing on the reward function outcome. The findings of the comparative study are summarized in Table 5.

With the Ensemble of Identical Independent Evaluators (EIIE) architecture, [64] an agent model for portfolio management (trading) based on ANN-based deep-learning (multi-layer ANN) has been developed. The data utilized comprised observations over 5283 days for 415 stocks in the S&P 500 stock market index, spanning twenty-one years from 1998 to the end of 2018. The reward function was based on Sharp and differential Sharpe ratios. Over a five-year test period, the model generated a trading policy that produced a return of 328.9% and a Sharpe ratio of 0.91. By optimizing stock trading strategies, Yang et al. [28] have created another deep reinforcement architecture model to increase investment return. Three actor–critic-based algorithms were used in the model: Proximal Policy Optimization (PPO), Advantage Actor Critic (A2C), and Deep Deterministic Policy Gradient (DDPG). Through use of this ensemble approach, it can adapt to changing market conditions by inheriting and incorporating the best characteristics of the three algorithms. A 7-year data set of 30 Dow Jones stocks was used to evaluate the model. The model performance resulted in a 70.4% cumulative return, Sharpe ratio of 1.3, and −9.7% maximum drawdown.

Théate and Ernst [65] have proposed the Trading Deep Q-Network algorithm (TDQN), a deep reinforcement learning (DRL) technique for determining the optimum trading position by maximizing the resultant Sharpe ratio performance. The model was evaluated by analysing 30 stocks from a variety of industries in North America, Europe, and Asia, in order to determine how well it performs. With an initial investment of $100,000, the TDQN algorithm achieved good results from both earnings and risk mitigation standpoints. In the case of Apple stock, for example, the execution of the TDQN trading strategy yielded a Sortino ratio of 1.84 with an annualised return of 32%. On Tesla stock, the performance reached 0.359 in Sortino ratio and an annualized return exceeding 12%. In general, the TDQN algorithm achieved an average Sharpe Ratio of 0.401 for all 30 stocks.

**Table 5.** Comparative Performance Analysis of Our Model with Other Models.

| Reference | Year | Data Set | Rewards | Performance |
|---|---|---|---|---|
| Huotari et al. [64] | 2020 | 415 stocks in the S&P 500 stock | Sharpe<br>Differential Sharpe ratios | return: 328.9%<br>Sharpe ratio: 0.91 |
| Yang et al. [28] | 2020 | 30 Dow Jones stocks | Portfolio value change Turbulence | return: 328.9%<br>Sharpe ratio: 0.91 |
| Théate and Ernst [65] | 2021 | 30 stocks from a variety of industries in North America, Europe, and Asia | Daily Returns | All 30 stocks<br>Sharpe Ratio: 0.401<br>APPL<br>Sortino ratio: 1.84<br>Annualised return: 32%<br>Tesla<br>Sortino ratio: 0.35<br>Annualised return:12% |
| Our Work | 2022 | 4 stocks from Saudi Stock Market (Tadawul) | Sortino ratio,<br>Cumulative Returns,<br>Annual Volatility | Net worth: 20%<br>Sortino ratio: 21.54%<br>Cumulative Returns: 14.5%<br>Annual Volatility: 11.1% |

## 6. Conclusions

In this research, we integrated modern transformer deep learning (TDL) into a traditional deep reinforcement learning (DRL) architecture for the processing of financial signals and automated trading. The transformer network in the proposed technique allowed the algorithm to predict the best trading strategy without looking back to track the price movements of stocks. Based on the data of four different industrial indices of the Saudi Stock Exchange (Tadawul), the proposed method provided the optimal learning of hyperparameters by automatic selection of the reward function, based on the maximum outcome of the network and return on investment. Of the seven considered reward functions, the Sortino ratio, cumulative returns, and annual volatility performed the best. The Sortino ratio provided the highest average increases in net worth for Saudi Telecom Company, Saudi Electricity Company, Saudi Basic Industries Corporation, and Al-Rajihi Banking and Investment (21.54%, 18.54%, 17%, and 19.36%, respectively); cumulative returns provided 8.02%, 10.61%, 12.92%, and 14.57% average increases in net worth for Saudi Telecom Company, Saudi Electricity Company, Saudi Basic Industries Corporation, and Al-Rajihi Banking and Investment, respectively; and annual volatility provided 7.84%, 11.1%, 5.92%, and 7.47% average increases in net worth for Saudi Telecom Company, Saudi Electricity Company, Saudi Basic Industries Corporation, and Al-Rajihi Banking and Investment, respectively. Considering the average increases in return and net worth, the proposed algorithm can be used in trading for long-term investment plans to gain profits.

In traditional deep reinforcement learning, agents are trained to optimize decisions to achieve the optimal return. At every time step, an agent observes the environment and decides which action to take to help itself achieve a higher return magnitude in future interactions. A Decision Transformer (DT), on the other hand, forecasts future behaviour by considering both intended future returns and previous interactions between the agent and its environment. By leveraging a causally masked Transformer, we produced optimal actions by mapping diverse experiences to their respective return magnitudes during training, as opposed to conventional RL, which computes policy gradients or fits value functions. Using a variety of experiences when training an agent increases the model's exposure to a wide range of trading variations, thereby helping it to derive useful trading rules that will enable it to succeed under any given circumstance. However, as Transformers scale quadratically with input size, the generation of optimal actions requires rather long DT response times. The implication of this approach naturally increases the time complexity; however, we anticipate that the parallelization of this approach on traditional or massively parallel architectures (e.g., GPUs) may allow for a reduction in the time complexity of our approach; this is expected to inform our future work.

**Author Contributions:** Conceptualization, N.M.; methodology, N.M.; software, N.M.; validation, N.M.; formal analysis, N.M. and R.M.; investigation, N.M., R.M. and I.K.; resources, R.M. and I.K.; data curation, N.M.; writing—original draft preparation, N.M.; writing—review and editing, R.M. and I.K.; visualization, N.M.; supervision, R.M. and I.K.; project administration, R.M. and I.K.; funding acquisition, None. All authors have read and agreed to the published version of the manuscript.

**Funding:** This research received no external funding.

**Data Availability Statement:** Not applicable.

**Acknowledgments:** The work carried out in this study was supported by the HPC Center at the King Abdulaziz University. The experiments reported in this paper were performed on the Aziz supercomputer at KAU. The authors are thankful to the anonymous reviewers whose comments helped us to significantly improve this paper.

**Conflicts of Interest:** The authors declare no conflict of interest.

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
