# Peer review of "Smart Robotic Strategies and Advice for Stock Trading Using Deep Transformer Reinforcement Learning"

_applsci, doi:10.3390/app122412526_

Round 1

Reviewer 1 Report

The article topic is still hot and interesting.

The proposed approached technically sounds.

The results are interesting. 

Author Response

Dear Reviewer,

Please see our response to your comments in the attached file. 

We have used the MDPI English Editing Services to improve the English in the paper. Please note that the tracked version of our paper and the text included in our authors' response file to you do not have these changes included. The clean version of the manuscript includes the English changes made to our paper. 

The clean version is submitted as the main file and the tracked version is submitted as the supplementary folder.

Thank you

Reviewer 2 Report

1 - The paper lacks the information about the main state space model. Model is not presented. There is some information given in the text, but there is no explaination as why the state vector is defined with 11 dimensions. No furhter modelling details are provided. 

2 - The paper is full of irrelevant information at different places. It can be shorten and presented with the core information about the model. 

3 - Its title is not appropriate and misleading as there is nothing related to robotics in the paper. It should be rewritten.   

4 - Information about the 4 companies is given at multiple places in the paper which can be removed. 

Author Response

(The authors gave the same response as above.)

Reviewer 3 Report

In this paper, they combine deep reinforcement learning-based (DRL) with transformer networks for online trading. They use data from the Saudi Stock Exchange (Tadawul), one of the largest liquid stock exchanges globally. Specifically, we use indices of four firms Saudi Telecom Company (7010), AlRajhi Banking & Investment (1120), Saudi Electricity Company (5110), and Saudi Basic Industries Corporation (2010). Some comments need to be resolved,

1. the abstract Reorganize the abstract to conclude: (a) The overall purpose of the paper and the research problems you investigated should be short. (b) The basic design of the study. (c) Major findings or trends found as a result of the study. (d) A brief summary of your interpretations and conclusions.

2. Many method improvements of the authors are the integration of a large number of existing methods. Are they simple system integration or have new features? How can the authors prove the effectiveness of the integration methods?

3. The accuracy of the proposed method is high, which is commendable. As everyone knows, increasing the complexity of the network can improve its accuracy. However, the lightweight network and reducing calculation time are also important parts and hot topics in the field of deep learning. What are the improvements of the proposed method in lightweight compared with classical network models? How do you balance the relationship between model accuracy and calculation speed?

4. Experiment section is weak. Please compare the proposed framework with other state-of-the-art methods.
5. The details of loss function are missing. Also provide details of training the model.

6. Discuss time complexity of the network.

7. Please improve the quality of figures as in the current form the images are blurry.

8. Please discuss the failure cases of proposed model and provide justification.

9. Author is advised to incorporate the recent research work published in this area. Compare and contrast your work with the latest algorithms and principles developed in the past few years.

Author Response

(The authors gave the same response as above.)

Round 2

Reviewer 2 Report

Now the state space model is comprised of 32 state vectors, changed from initial 11. There are details provided in 3.1.4 and Section 3.3, however, the model is not presented in the graphical form. This is understandable, that with so many state vectors, this is challenging, although, this may be shown in parts.

Author Response

Dear Reviewer,

Reviewer 3 Report

The response is clear.

Author Response

Dear Reviewer,
